# A new early branching armored dinosaur from the Lower Jurassic of southwestern China

Xi Yao[1†], Paul M Barrett[2†], Lei Yang[3], Xing Xu[1,4,5*], Shundong Bi[1,6*]

[1]Centre for Vertebrate Evolutionary Biology, Yunnan University, Kunming, China; [2]Department of Earth Sciences, Natural History Museum, London, United Kingdom; [3]Yimen Administration of Cultural Heritage, Yimen, China; [4]Key Laboratory of Evolutionary Systematics of Vertebrates, Institute of Vertebrate Paleontology and Paleoanthropology, Chinese Academy of Sciences, Beijing, China; [5]Center for Excellence in Life and Paleoenvironment, Beijing, China; [6]Department of Biology, Indiana University of Pennsylvania, Pennsylvania, United States

*For correspondence:
xu.xing@ivpp.ac.cn (XX);
sbi@iup.edu (SB)

[†]These authors contributed equally to this work

Competing interest: The authors declare that no competing interests exist.

**Abstract** The early evolutionary history of the armored dinosaurs (Thyreophora) is obscured by their patchily distributed fossil record and by conflicting views on the relationships of Early Jurassic taxa. Here, we describe an early diverging thyreophoran from the Lower Jurassic Fengjiahe Formation of Yunnan Province, China, on the basis of an associated partial skeleton that includes skull, axial, limb, and armor elements. It can be diagnosed as a new taxon based on numerous cranial and postcranial autapomorphies and is further distinguished from all other thyreophorans by a unique combination of character states. Although the robust postcranium is similar to that of more deeply nested ankylosaurs and stegosaurs, phylogenetic analysis recovers it as either the sister taxon of *Emausaurus* or of the clade *Scelidosaurus*+ Eurypoda. This new taxon, *Yuxisaurus kopchicki*, represents the first valid thyreophoran dinosaur to be described from the Early Jurassic of Asia and confirms the rapid geographic spread and diversification of the clade after its first appearance in the Hettangian. Its heavy build and distinctive armor also hint at previously unrealized morphological diversity early in the clade's history.

## Editor's evaluation

This paper reports a new species of armored dinosaur from rocks in southwestern China dated to the beginning of the Jurassic Period. This represents the first valid species of armored dinosaur from the Early Jurassic in Asia, as although the presence of armored dinosaurs in Asia has been documented for decades based on isolated jaw bones referred to Thyreophora-the group of armored dinosaurs-none that material was complete enough for diagnosis to a known or new species. This new specimen demonstrates the rapid diversification and distribution of armored dinosaurs across the northern hemisphere early in their evolutionary history.

## Introduction

Thyreophoran dinosaurs were important components of many terrestrial faunas from the Late Jurassic until the end of the Cretaceous, particularly in Laurasia (*Vickaryous and Russell, 2003*; *Galton, 2004*; *Arbour and Currie, 2015*; *Maidment et al., 2020*). However, many aspects of their earlier evolutionary history remain contentious and poorly known. The majority of late Mesozoic armored dinosaurs belonged to one of two major lineages—Ankylosauria or Stegosauria—whose earliest members

**eLife digest** From the plated *Stegosaurus* to the tank-like *Ankylosaurus*, armoured dinosaurs are some of the most extraordinary creatures to have roamed the earth. Fossils from this group are abundant from the Late Jurassic period, 155 million years ago, up until the end of the age of the dinosaurs. However, only a few fossils exist from the early part of the Jurassic, making it difficult to understand how these fantastic beasts came to be. More early fossils could help to fill in gaps about armoured dinosaur biology and evolution.

Yao et al. describe the anatomy of a new armoured dinosaur, baptized *Yuxisaurus*, which was found in rocks of Early Jurassic age in southwestern China. Covered in sharp spines, this medium-sized animal was much sturdier and stockier than its immediate relatives, suggesting that the ancestors of *Stegosaurus* and *Ankylosaurus* had a wider variety of body forms than once thought. Its presence in China also shows that armoured dinosaurs spread across the world early in their history. *Yuxisaurus* could help researchers to understand how million years of evolution produced the armoured species we are more familiar with today. As more fossils may emerge from the rocks of southwestern China, it could become possible to further piece together early dinosaur evolution.

are currently known from the Middle Jurassic (*Galton, 1983*; *Salgado et al., 2017*; *Maidment et al., 2020*; *Maidment et al., 2021*). Almost all recent analyses of ornithischian interrelationships have united these two lineages in a clade named Eurypoda, which is thought to have originated sometime in the Early–early Middle Jurassic (e.g., *Sereno, 1999*; *Norman et al., 2004*; *Butler et al., 2008*; *Boyd, 2015*; *Dieudonné et al., 2021*).

However, several possible Early Jurassic thyreophorans lack key ankylosaurian and stegosaurian synapomorphies. These include *Laquintasaura* and *Lesothosaurus*, which are recovered as early, unarmored thyreophorans by some phylogenetic analyses (*Butler et al., 2008*; *Boyd, 2015*; *Baron et al., 2017a*) but placed in alternative positions outside Thyreophora in others (*Sereno, 1999*; *Dieudonné et al., 2021*). Less controversially, three other taxa are consistently recovered as early diverging members of the clade: *Scutellosaurus lawleri* (Sinemurian–Toarcian, Kayenta Formation, USA; *Colbert, 1981*; *Rosenbaum and Padian, 2000*; *Breeden and Rowe, 2020*; *Breeden et al., 2021*), *Emausaurus ernsti* (early Toarcian, unnamed unit, Germany; *Haubold, 1990*), and *Scelidosaurus harrisonii* (Sinemurian–early Pliensbachian, Charmouth Mudstone Formation, UK; *Owen, 1861*; *Owen, 1863*; *Norman, 2020a*; *Norman, 2020b*; *Norman, 2020c*).

Most recent studies have concluded that *Scutellosaurus*, *Emausaurus,* and *Scelidosaurus* are successive sister taxa to Eurypoda (*Sereno, 1999*; *Norman et al., 2004*; *Butler et al., 2008*; *Boyd, 2015*; *Dieudonné et al., 2021*). However, an alternative hypothesis suggests that *Scelidosaurus* was the sister taxon of Ankylosauria, together forming the clade Ankylosauromorpha, which in turn is the sister group of Stegosauria. This relationship was first proposed formally by *Carpenter, 2001* and received support from *Norman, 2021*; but see Results, below. Testing these alternatives will rely on the discovery of new material and on the construction of larger phylogenetic data matrices including more characters suited to unraveling early thyreophoran relationships.

Two probable thyreophoran taxa have been described from the Early Jurassic of China—'*Bienosaurus lufengensis*' and '*Tatisaurus oehleri*'—both erected on the basis of fragmentary material from the Lower Jurassic Lufeng Formation of Yunnan Province (*Simmons, 1965*; *Dong, 2001*). However, in both cases, the material is insufficient to support their validity and these taxa are currently regarded as nomina dubia, although the material does exhibit thyreophoran characteristics (*Norman et al., 2007*; *Raven et al., 2019*). Consequently, these specimens offer little useful information on thyreophoran evolution, although they do extend the range of the clade to East Asia at this time, suggesting that the group achieved a global (or at least pan-Laurasian) distribution soon after its origin (*Raven et al., 2019*).

Here, we describe a new thyreophoran taxon from the Lower Jurassic Fengjiahe Formation of Yunnan Province, southwestern China on the basis of a partial skeleton and discuss its significance for early ornithischian evolution.

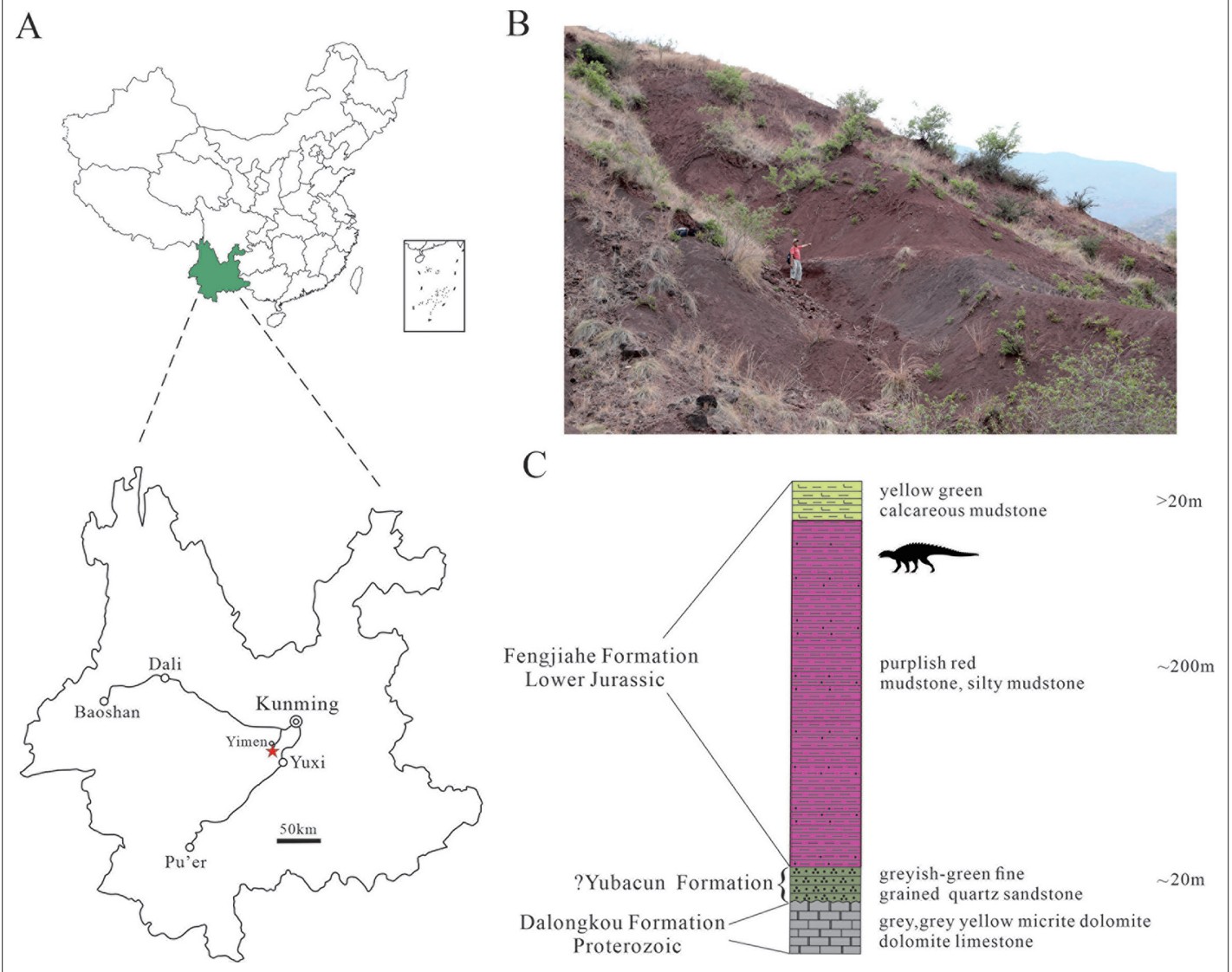

**Figure 1.** Geographical and stratigraphic location of *Yuxisaurus kopchicki* CVEB21701. (**A**) Location of the quarry yielding *Yuxisaurus kopchicki*, with a red star indicating the locality. (**B**) Sediments of the Fengjiahe Formation at the quarry site. (**C**) Stratigraphic column of the Fengjiahe Formation in the Jiaojiadian area (modified from *Bai, 1999*).

## Geological setting

The main exposures of the Fengjiahe Formation are found in the Chuxiong Basin and Yiliang region of central and northeastern Yunnan, respectively (*Figure 1A and B*). It consists primarily of dull purplish and dark red mudstone and siltstone, mixed with yellowish or greyish green siltstone and quartz sandstone, calcareous mudstone, and nodules (*Fang et al., 2008*). *Pang et al., 2002* recognized a transition bed between the underlying coal-bearing Shezi Formation and the overlying Fengjiahe Formation and designated this transitional bed as a new lithostratigraphic unit, the Yubacun Formation. This revision resulted in the separation of the lower variegated beds from the overlying purple sediments of the Fengjiahe Formation (*Pang et al., 2002*). Although the presence of the Yubacun Formation in the Jiaojiadian area has not been confirmed, the lower greyish-green sandstones formerly referred to as the Fengjiahe Formation in this area coincide well with the lithology of the Yubacun Formation and are now considered to represent this unit (*Figure 1C*). Here, therefore, we restrict the Fengjiahe Formation to the sequence above these greyish-green sandstones (*Figure 1C*).

The Fengjiahe Formation is currently thought to be a lateral equivalent of the Lufeng Formation, which crops out in the adjacent Lufeng Basin (*Fang et al., 2008*). Biostratigraphical correlations

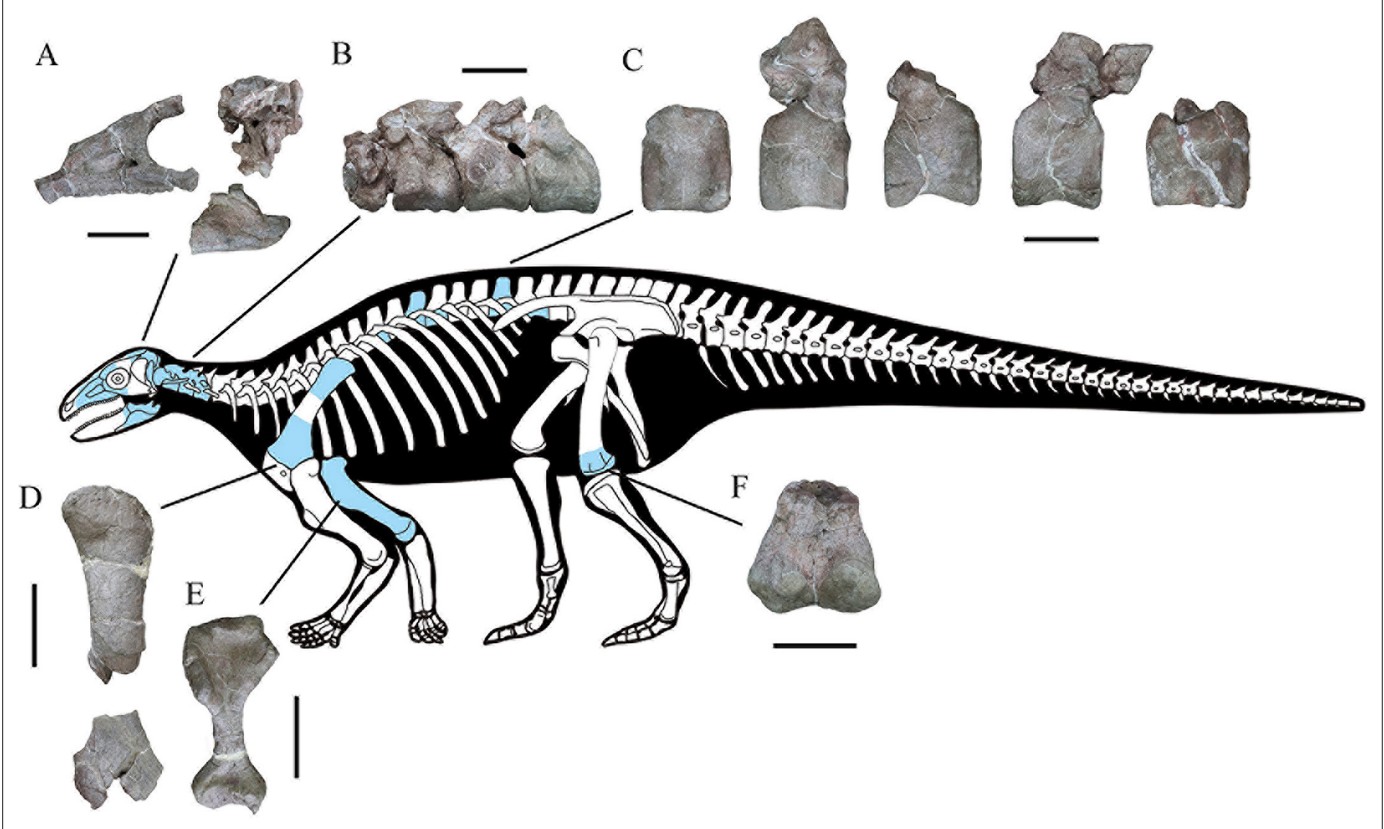

**Figure 2.** Skeletal reconstruction of *Yuxisaurus kopchicki* showing some of the main preserved elements from the holotype (highlighted in blue), with details of the skull bones (**A**), cervical vertebrae (**B**), dorsal vertebrae (**C**), left scapula (**D**), right humerus (**E**), and left femur (**F**). Scale bars equal 5 cm (**A–C**) or 10 cm (**D–F**). The facial region and distal scapula are mirrored. Osteoderms have been omitted for convenience.

based on fossil vertebrates have suggested that the Lufeng Formation is Lower Jurassic (Hettangian–Sinemurian) in age (*Luo and Wu, 1994*), and the similar vertebrate fauna and correlations based on invertebrate and micropaleontological material from the Fengjiahe Formation are consistent with this (*Chen et al., 1982*). However, more recent magnetostratigraphic evidence posits a younger age for the Lufeng Formation, namely late Sinemurian–Toarcian (*Huang et al., 2005*). Although it has not yielded as many vertebrate fossils as the Lufeng Formation, the Fengjiahe Formation has produced several important early sauropodomorph dinosaurs, such as *Chinshakiangosaurus chunghoensis*, *Irisosaurus yimenensis*, *Lufengosaurus huenei*, *Yunnanosaurus huangi*, *Y. robustus*, and *Yimenosaurus yangi*, as well as the theropod *Shuangbaisaurus anlongbaoensis* and dinosaur footprints (*Zhen et al., 1986*; *Bai et al., 1990*; *Bai, 1999*; *Dong, 2001*; *Upchurch et al., 2007*; *Wang et al., 2017*; *Peyre de Fabrègues et al., 2020*). The new thyreophoran was discovered in the upper part of the Fengjiahe Formation, as is usually the case for the vertebrate material recovered from this stratum.

## Results

### Systematic paleontology

Dinosauria *Owen, 1842*
Ornithischia *Seeley, 1887*
Thyreophora *Nopcsa, 1915* (sensu *Sereno, 1998*)
*Yuxisaurus kopchicki* gen. et sp. nov.
urn:lsid:zoobank.org:pub:6C8204FE-1A51-4E7D-B9B5-BE9939460D6E

## Holotype

CVEB (Centre for Vertebrate Evolutionary Biology, Yunnan University) 21,701 a partial skeleton with cranial and associated postcranial elements (*Figure 2*), including: the right-hand side of the skull (fused maxilla, lacrimal, nasal, prefrontal, jugal, and supraorbitals); braincase; partial skull roof; posterior parts of the hemimandibles; four articulated cervical vertebrae; five dorsal vertebrae; left proximal and right distal scapulae; right humerus; left distal femur; more than 120 osteoderms; and several unidentifiable elements.

## Etymology

The generic name refers to the type locality in Yuxi Prefecture, with the suffix -saurus from the Greek, meaning reptile. The specific name is after Dr. John J. Kopchick in recognition of his contributions to biology and the IUP Science Building.

## Horizon and locality

Upper part of the Fengjiahe Formation, near Jiaojiadian village, Yimen County, Yuxi Prefecture, Yunnan Province, China; late Sinemurian–Toarcian (*Huang et al., 2005*; *Figure 1C*).

## Diagnosis

A medium-sized armored dinosaur that can be distinguished from all other thyreophorans by the following autapomorphies: deep, subtriangular, dorsoventrally elongated depression on either side of the nuchal crest; a 'V'-shaped notch on the dorsal margin of the paroccipital process; basal tubera that are considerably ventrally offset with respect to the occipital condyle, so that they are clearly visible in posterior view; basipterygoid processes that are ventrally offset with respect to the basal tubera, creating a dorsoventrally deep, 'stepped' basicranial profile in lateral view; cultriform process ventrally offset with respect to the occipital condyle in lateral view; angular with elongate, dorsally deflected posterior process that almost reaches the posterior margin of the retroarticular process; atlas intercentrum with symmetrical anterolaterally directed low ridges and associated arrow-like depressions on its ventral surface; relatively short anterior cervical centra (length/height ratio<1.5); cervical centra lack ventral keels; border of the medial condyle of the distal femur invaginated to form a broad, 'U'-shaped trough.

In addition, *Yuxisaurus* can be distinguished from other early thyreophorans using the following combination of character states: antorbital fossa subtriangular in outline, unlike that of *Scelidosaurus*, and with rounded corners, unlike that in *Scutellosaurus*; anterior ramus of the jugal projects posteroventrally, rather than horizontally as in *Emausaurus*, *Scelidosaurus,* and *Scutellosaurus*; maxillary tooth row bowed medially to a greater degree than in *Emausaurus*, *Scelidosaurus,* or *Scutellosaurus*; maxillary tooth crowns bearing well-defined ridges, which are absent in *Emausaurus*, *Scelidosaurus,* and *Scutellosaurus*; a relatively short axial neural spine with a sinuous dorsal margin in lateral view, contrasting with the straight margin and significant posterior expansion of the neural spine present in *Scelidosaurus*; elongate axial rib, which extends to the midpoint of cervical vertebra 3, unlike the shorter rib present in *Scelidosaurus*; absence of lateral ridge on the axial rib, which is present in *Scelidosaurus*; proximal and distal expansions of the humerus relatively larger than in *Scelidosaurus* and *Scutellosaurus*; deep notch separating the humeral head and dorsal margin of the internal tuberosity, which is absent in *Scelidosaurus* and *Scutellosaurus*; and broad, 'U'-shaped fossa on anterior surface of distal humerus, contrasting with the narrow, 'V'-shaped fossae in *Scelidosaurus* and *Scutellosaurus*.

## Remarks

The other thyreophoran taxa named from the Early Jurassic of China ('*Bienosaurus*' and '*Tatisaurus*') are based on undiagnostic material (*Norman et al., 2007*; *Raven et al., 2019*) and have limited anatomical overlap with *Yuxisaurus*. Consequently, it is not possible to make meaningful comparisons between them and no shared features can be identified. As a result, additional specimens will be required to establish whether these three named taxa are synonymous or if multiple thyreophoran taxa were present in the Early Jurassic of China.

## Description and comparisons
### General comments
The cranial bones are highly fused and the neurocentral sutures of all preserved cervical and dorsal vertebrae are invisible, in particular the completely obliterated axial neurocentral suture, suggesting that this specimen might represent an adult individual (*Brochu, 1996*). Compared to other closely-related taxa, the skull of *Yuxisaurus kopchicki* is larger than those of *S. lawleri* (*Breeden and Rowe, 2020*; *Breeden et al., 2021*), *E. ernsti* (*Haubold, 1990*), and *S. harrisonii* (Natural History Museum, London [NHMUK] PV R1111; *Norman, 2020c*), and *Y. kopchicki* has much more robust fore- and hind-limbs than the latter.

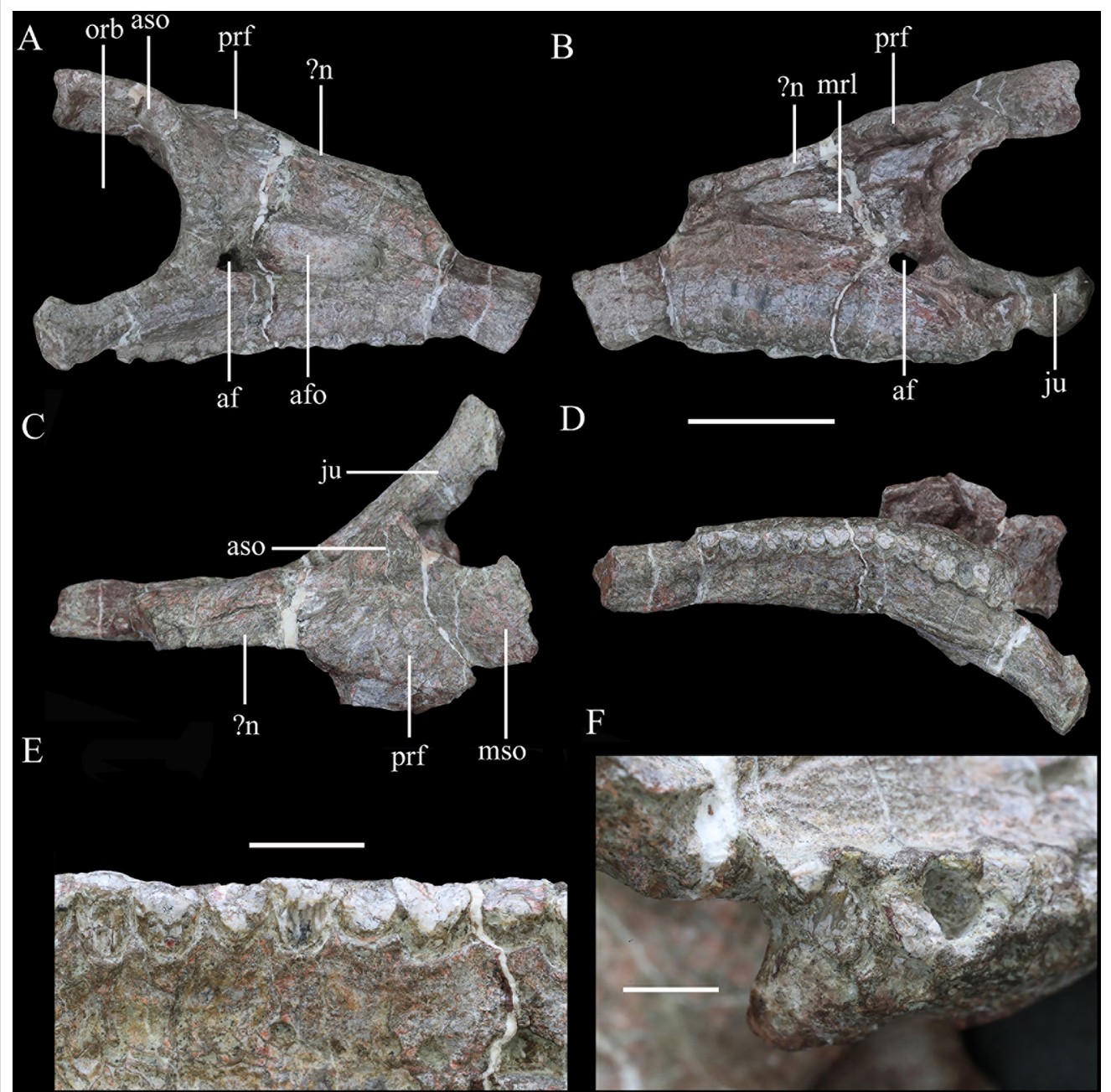

**Figure 3.** Right maxilla of *Yuxisaurus kopchicki* in (**A**) lateral, (**B**) medial, (**C**) dorsal, and (**D**) ventral views. Maxillary tooth row in (**E**) lingual view with the last tooth in (**F**) lingual view. Abbreviations: af, antorbital fenestra; afo, antorbital fossa; fenestra; aso, anterior supraorbital; ju, jugal; mrl, maxillary ramus of the lacrimal; mso, mesosupraorbital; orb, orbital; aso anterior supraorbital; prf, prefrontal. Scale bar equals 5 cm.

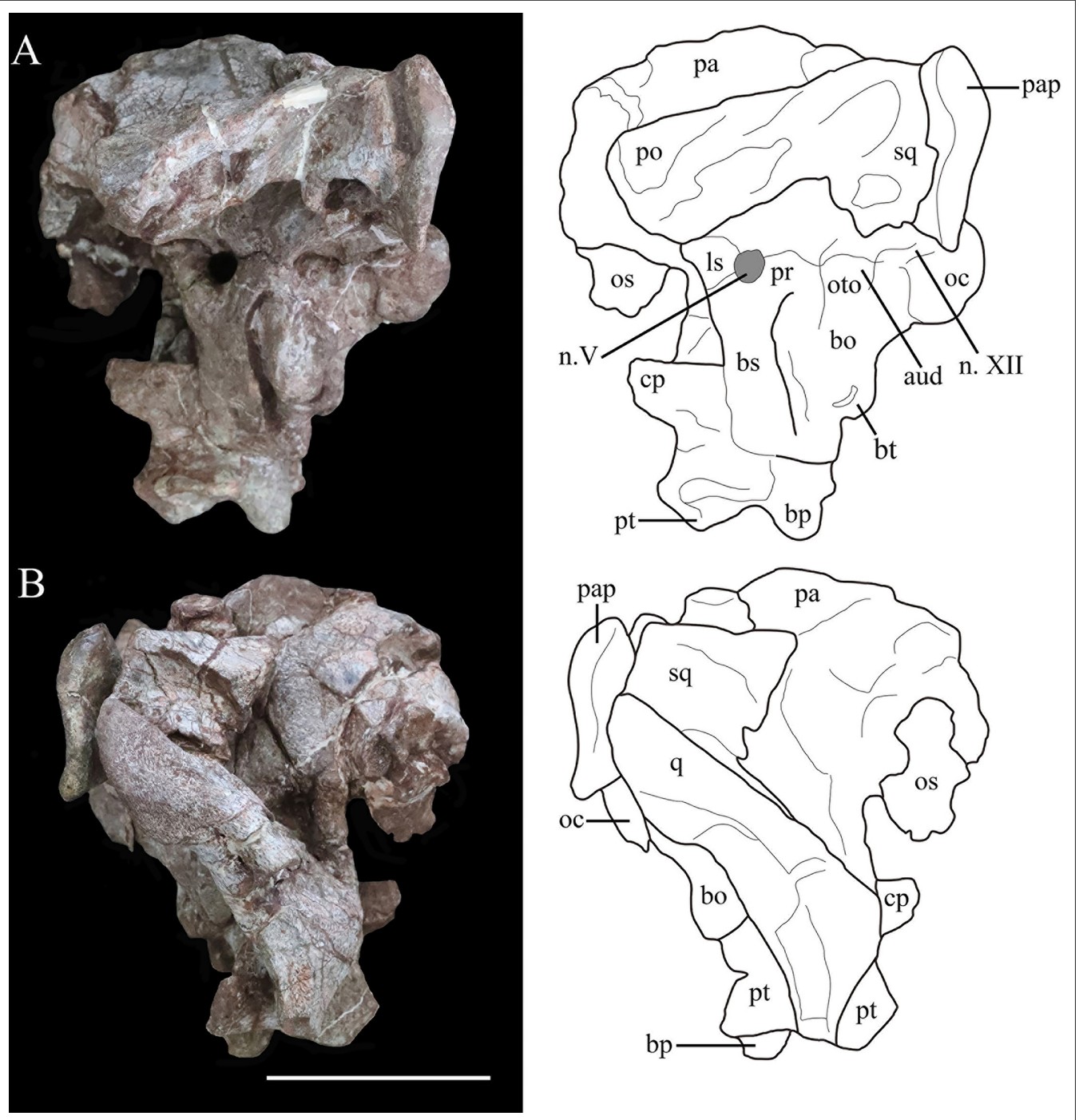

**Figure 4.** Photographs (left) and line drawings (right) of the braincase and partial skull roof of *Yuxisaurus kopchicki* in left lateral (**A**) and right lateral (**B**) views. Abbreviations: aud, auditory recess; bo, basioccipital; bp, basipterygoid process; bs, basisphenoid; cp, cultriform process (parasphenoid rostrum); fm, foramen magnum; ls, laterosphenoid; n. V, exit of trigeminal nerve; n. XII, exit of cranial nerve XII; oc, occipital condyle; os, orbitosphenoid; oto, otoccipital; pa, parietal; pap, paroccipital process; po, postorbital; pr, prootic; pt, pterygoid; q, quadrate; sq, squamosal. Scale bar equals 5 cm.

## Skull and mandible

The skull includes a braincase, part of the skull roof, the co-ossified right side of the facial region (including the maxilla, anterior and mesosupraorbitals, lacrimal, prefrontal, jugal, and probable nasal), and the posterior parts of both hemimandibles (*Figure 2*, *Figure 3*, *Figure 4*, *Figure 5*, *Figure 6*, *Figure 7*).

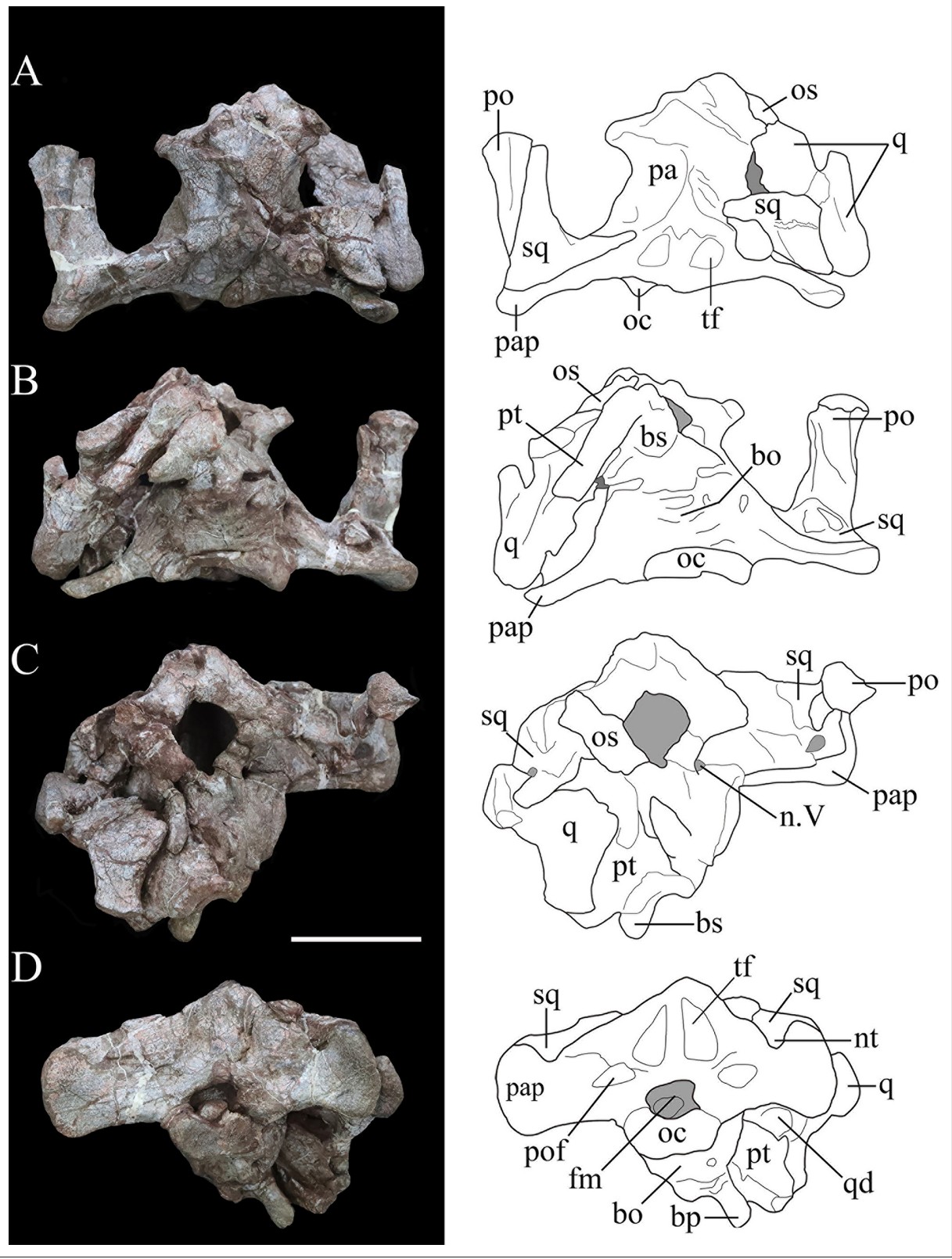

**Figure 5.** Photographs (left) and line drawings (right) of the braincase of *Yuxisaurus kopchicki* in (**A**) dorsal, (**B**) ventral, (**C**) anterior, and (**D**) posterior views. Abbreviations: bo, basioccipital; bp, basipterygoid process; bs, basisphenoid; cp, cultriform process (parasphenoid rostrum); fm, foramen magnum; nt, 'V'-shaped notch; n. V, exit of trigeminal nerve; oc, occipital condyle; os, orbitosphenoid; pa, parietal; pap, paroccipital process; po, postorbital; pt, pterygoid; q, quadrate; qd, quadrate depression; sq, squamosal. Scale bar equals 5 cm.

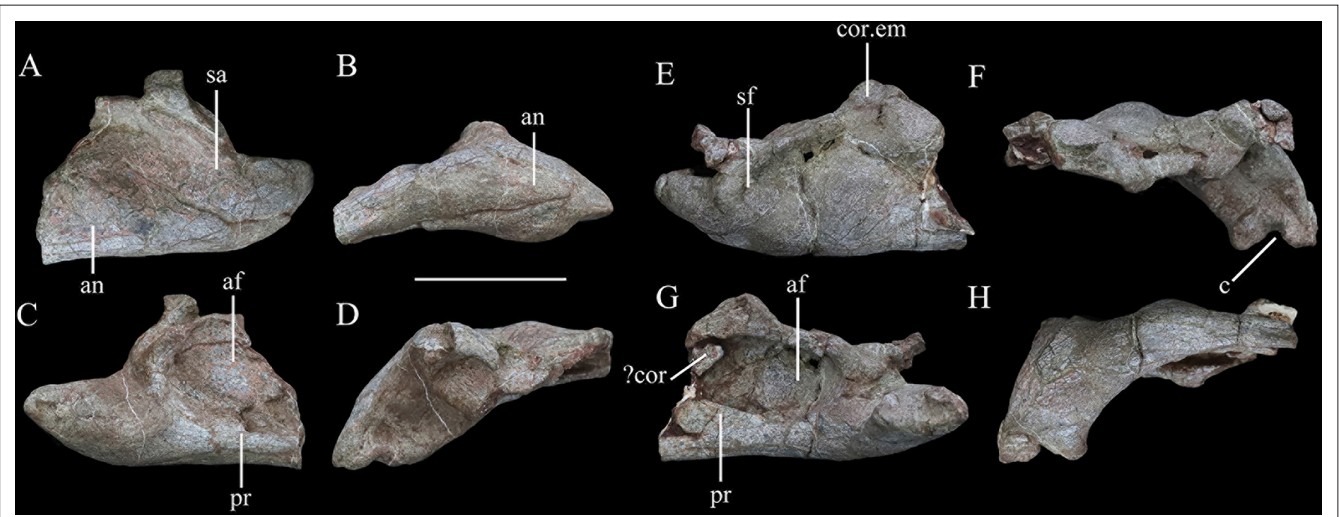

**Figure 6.** Possible skull roof fragment of *Yuxisaurus kopchicki* in (**A**) dorsal, (**B**) lateral, and (**C**) ventral views. Abbreviations: cd, channel-like depression; d, dome. Scale bar equals 5 cm.

## Maxilla

In lateral view, the right maxilla appears to be fused completely with the jugal posteriorly, the lacrimal posterodorsally, and the nasal medially, with few identifiable sutures. Its anterior part is broken. An anteroposteriorly elongated antorbital fossa excavates its lateral surface deeply. The antorbital fossa is rounded and subtriangular in outline with long anterodorsal and ventral margins and a short postero-dorsal margin (*Figure 3A*). The fossa reaches a maximum length of 48 mm and is 20 mm in height at its apex. Most of the antorbital fossa is closed medially by an extensive, sheet-like medial lamina, but a small, oval antorbital fenestra pierces its posteroventral corner (*Figure 3A*). This region differs from that of *Scelidosaurus*, which has a relatively smaller antorbital fossa with a dorsoventrally narrow, elliptical outline (NHMUK PV R1111; *Norman, 2020c*), but is very similar to that of *Emausaurus* (*Haubold, 1990*). It differs from those of *Lesothosaurus* (e.g., NHMUK PV RU B17; *Porro et al., 2015*), *Scutellosaurus* (*Breeden and Rowe, 2020*), and *Huayangosaurus* (*Sereno and Dong, 1992*) in having a fossa with smooth, rounded corners, in contrast to the sharp, angular corners seen in the latter taxa. *Yuxisaurus* also appears to lack the anterior antorbital foramen present in *Scelidosaurus* (*Norman, 2020c*), but this area is still encased in matrix.

Ventral to the antorbital fossa is the medially inset buccal emargination, which is approximately 30-mm tall along most of its length except where the alveolar margin curves dorsally at its posterior end (*Figure 3A*). The buccal emargination is generally smooth and mildly depressed and contains several, small irregularly placed shallow depressions that might have been caused by weathering. The dorsal boundary of the buccal emargination is formed by a distinct, rounded ridge. Dorsal to this ridge, most of the lateral surface of the right maxilla is slightly convex, although the part anterior to the antorbital fossa is flat. The alveolar margin is scalloped in lateral view.

**Figure 7.** Mandibular remains of *Yuxisaurus kopchicki*. Posterior part of left hemimandible in (**A**) lateral, (**B**) ventral, (**C**) medial, and (**D**) dorsal views. Posterior part of right hemimandible in (**E**) lateral, (**F**) ventral, (**G**) medial, and (**H**) dorsal views. Abbreviations: af, adductor fossa; an, angular; c, concavity; cor, coronoid; cor.em, coronoid eminence; pr, prearticular; sa, surangular; sf, surangular foramen. Scale bar equals 5 cm.

In medial view, a series of small rounded replacement foramina, which correspond one-to-one with the alveolar sockets, lies immediately above the alveolar margin (*Figure 3B*). The rest of the surface dorsal to the alveolar margin is smoothly convex, producing a vertical flange that extends dorsally for a short distance. The dorsal margin of this flange bears a shallow horizontal trough, which curves laterally anteriorly as well as posteriorly to communicate with the antorbital fenestra. It then continues posteroventrally for 21 mm (*Figure 3B*). The dorsal boundary of the flange is straight and oblique anteriorly, but curves downward posteriorly. Another groove starting halfway along the above-mentioned trough extends posteriorly and expands into an elongated deep sulcus (*Figure 3B and C*). This groove probably represents the articular contact between the maxilla and the lacrimal/jugal. The bone sandwiched between these two grooves has a dorsal concavity terminating posteriorly in a blunt process, which grades into the deep fossa mentioned above.

The antorbital fenestra is a rounded opening in medial view. The medial (lacrimal) lamina of the right maxilla is concealed medially by the anterior (medial) process of the lacrimal. The articulation between the lacrimal and maxilla is clear anteriorly but indistinguishable posteriorly. The posterior part of the medial surface dorsal to the tooth row is sculptured, probably indicating the contact surface with the palatine.

In ventral view, the alveolar border is bowed medially and the deflection angle between the anterior and posterior axes of the tooth row is approximately 148° (*Figure 3D*). The curvature in *Yuxisaurus* is not as extreme as that present in many ankylosaurians where the tooth row is strongly bowed (*Vickaryous and Russell, 2003*), but is greater than that in *Scelidosaurus*, *Emausaurus*, *Scutellosaurus,* and stegosaurians in which the maxillary tooth row is almost straight and only slightly curved (*Colbert, 1981*; *Haubold, 1990*; *Sereno and Dong, 1992*; *Galton, 2004*; *Breeden and Rowe, 2020*; *Norman, 2020c*; *Breeden et al., 2021*). Based on the number of alveoli present (*Figure 3D*), *Yuxisaurus* possessed at least 14 maxillary teeth.

## Lacrimal
The lacrimal lacks any discernible sutures with the surrounding bones except medially (part of its junction with the maxilla; see above) and with the anterior supraorbital ( = palpebral) (where a curved groove might mark the boundary) (*Figure 3A–C*). Based on comparisons with other thyreophorans, the lacrimal is inferred to comprise the anterior margin of the orbit and to contact the maxilla anteriorly and ventrally, the jugal posteroventrally, and the anterior supraorbital and prefrontal dorsally and posteriorly. The lateral surface of the lacrimal is sculptured and rugose, particularly in the region of the orbital margin. Its posterior surface (i.e., the anterior margin of the orbit) is concave and rounded in lateral view. In the border of the orbit, a rounded opening is present, indicating the exit of the nasolacrimal duct. The posterior margin of the lacrimal expands medially, to form a partition that separates the orbit from the nasal cavity anteriorly (*Figure 3B*). In medial view, this wall becomes thinner as it curves dorsally and slightly posteriorly to approach the prefrontal. The maxillary ramus of the lacrimal is an anteriorly trending triangular lamina that is concave in medial view, tapering at its anterior end. Due to the absence of recognizable sutures, it is not possible to determine the extent of the lacrimal's contribution to the antorbital fossa and fenestra.

## Nasal
A small fragment of bone anterior to the right prefrontal might represent part of the right nasal (*Figure 3A–C*). However, it cannot be identified with confidence and offers no useful information.

## Prefrontal
The right prefrontal roofs the nasal cavity dorsally (*Figure 3A*) and is flat ventrally but slightly domed in dorsal view (*Figure 3C*). It contacts the mesosupraorbital laterally and the middle supraorbital posteriorly. The prefrontal probably contacts the lacrimal anteriorly but this cannot be substantiated due to lack of a clear suture. A fractured bone anterior to the prefrontal, medial to the maxilla, probably belongs to the right nasal (see above).

## Supraorbitals
The anterior supraorbital ( = palpebral of other ornithischians) is represented by its anterior part only, which occupies the upper boundary of the orbit (*Figure 3A*). The anterior supraorbital is a narrow, elongated bone, which is co-ossified with the lacrimal anteroventrally, the prefrontal anteromedially,

and the mesosupraorbital medially. Viewed laterally, the anterior supraorbital curves posterodorsally from the anterodorsal margin of the orbit (*Figure 3A*). In dorsal view, it has a rounded anterior end to contact the lacrimal, while its contact with the mesosupraorbital is unclear. On the dorsal surface of the anterior supraorbital a distinct ridge extends posterodorsally (*Figure 3C*). The mesosupraorbital is partly preserved. It bulges dorsally but is concave ventrally and contacts the anterior supraorbital anteriorly through an anterolateral-posteromedial directed suture that turns into a groove dorsally.

### Jugal

The partly preserved right jugal articulates with the maxilla and lacrimal anteriorly. In lateral or medial view, the anterior ramus of the jugal projects posteriorly and slightly ventrally, whereas in dorsal or ventral view it extends posterolaterally (*Figure 3A–D*). *Yuxisaurus* differs from *Emausaurus*, *Scutellosaurus,* and *Scelidosaurus*, in which the anterior ramus is oriented horizontally (*Haubold, 1990*; *Breeden and Rowe, 2020*; *Norman, 2020c*; *Breeden et al., 2021*), but is more similar to several ankylosaurians, such as *Pinacosaurus*, *Gobisaurus*, *Saichania,* and *Edmontonia*, where the anterior ramus projects posteroventrally (*Godefroit et al., 1999*; *Vickaryous et al., 2001*; *Vickaryous, 2006*; *Carpenter et al., 2011*). The transverse cross-section of the jugal anterior ramus is rhomboidal but its posterior end is transversely compressed and dorsoventrally expanded. The posteromedial margin is inverted, leaving a dorsoventrally oriented embayment exposed in medial view.

### Postorbital

The postorbital is represented only by the left squamosal process, which formed part of the supratemporal bar. This process is bullet-shaped in dorsal view with a wide anterior end and pointed posterior end (*Figure 4A and B*). It is rhomboidal in cross-section with a flat dorsal surface that lies lateral and dorsal to the squamosal. The postorbital formed part of the dorsal margin of the infratemporal fenestra, but no other details are visible.

### Squamosal

The right squamosal is broken anteriorly and is slightly displaced medially, while the left squamosal articulates with the squamosal process of the left postorbital (*Figures 4A, B, 5A and B*). The squamosal is broad posteriorly, tapers anteriorly, and the dorsal surface of its central body is flat (*Figures 4B and 5A*). Its anterodorsal process is about 35-mm long and extends anteriorly and a little ventromedially, so that in dorsal view this process lies both medial and ventral to the squamosal process of the postorbital. In ventral view, this process is transversely narrow. The left anteroventral process is missing but this feature is preserved on the right side. It is rod-like but truncated anteriorly, and its dorsal part encloses a deep oval sulcus on the lateral surface (*Figure 4A and B*). The posteromedial process is dorsoventrally tall, merging with the squamosal process of the parietal posteriorly without a discernible suture on the posterior wall of the supratemporal fenestra. In medial view, at the base of the squamosal central body, is a fossa that is much broader on the right side than on the left. In lateral view, a similar but deeper recess is situated at the base of the squamosal central body to receive the quadrate head (*Figure 4A and B*). Posteriorly a short vertical process of the squamosal abuts the anterior surface of the paroccipital process (*Figure 4*). Viewed posteriorly, the squamosal is exposed dorsally, but it is positioned only slightly higher than the paroccipital process, as also occurs in *Lesothosaurus* (*Sereno, 1991*). By contrast, the squamosal has a much greater exposure in posterior view in *Scelidosaurus* and ankylosaurians (*Vickaryous and Russell, 2003*; *Norman, 2020c*), although the degree of exposure varies among stegosaurs (*Gilmore, 1914*; *Sereno and Dong, 1992*).

In dorsal view, the squamosal forms most of the medial margin of the large supratemporal fenestra, as well as its posterior corner. Although the boundaries of neither supratemporal fenestra are complete, the preserved portion on the left-hand side of the skull suggests that it had an ovate to subtriangular outline similar to that of *Emausaurus* (*Haubold, 1990*) and *Scelidosaurus* (NHMUK PV R1111; *Norman, 2020c*). The squamosal also formed the posterodorsal corner of an open infratemporal fenestra (*Figure 5A*).

### Quadrate

The right quadrate is partially preserved with its ventral-most part missing and the quadrate head displaced from the squamosal recess. In lateral view, the posterior margin of the quadrate is sinuous, being convex in its dorsal part but inflected at a point around one-third of its length so that ventral to

this the rest of this margin is shallowly concave (*Figure 4B*). In posterior view, the proximal quadrate bears a strong, curved crest. Although the ventral part is missing, it seems to curve ventromedially based on the remaining shaft. The pterygoid wing is laminar and extends anteromedially from the middle of the shaft to meet the quadrate ramus of the pterygoid (*Figures 4B and 5B*). A large oval depression occupies the medial surface of the pterygoid wing, as in *Scelidosaurus* (*Norman, 2020c*).

### Parietal

The parietal fuses with its counterpart to form an hourglass-shaped compound bone in dorsal view that bears a prominent sagittal crest (*Figure 5A*). The parietal fuses fully with the laterosphenoid anteroventrally and the prootic posteroventrally with no traceable boundaries between them. The posterior portion of the right parietal is damaged. The smooth lateral surfaces are concave antero-posteriorly but convex transversely and curve outward to form a short anterolateral process. In lateral view, the parietal extends to a level much higher than the squamosal (*Figure 4B*), in contrast to *Scelidosaurus* and stegosaurs in which the parietal is either only slightly elevated or at the same level (*Gilmore, 1914*; *Sereno and Dong, 1992*; *Norman, 2020c*). A deep sulcus is present on the main body of the left parietal close to the junction between the left medial and posterior supratemporal walls (*Figure 5A*), but this is not visible on the right-hand side, where it is concealed by the displaced squamosal. The parietal forms the medial boundary of the open supratemporal fenestra.

### Pterygoid

The pterygoid is partially preserved on the right side and is situated between the quadrate and the basipterygoid process of the basisphenoid. In posterior view, its quadrate ramus is a fan-shaped lamina that extends laterodorsally to meet the pterygoid wing of the quadrate (*Figure 5D*). Its ventral margin curls dorsally to form a narrow trough that is visible in posterior view as in *Lesothosaurus* and *Scelidosaurus* (*Sereno, 1991*; *Norman, 2020c*).

### Skull roof fragment

A broken plate-like element is tentatively identified as part of the skull roof, but it is unclear how it relates to the other cranial elements (*Figure 6*). Its most conspicuous feature is its wave-like surface texture, which is due to its domed external surface combined with the presence of a channel-like depression. This feature might be unique to *Yuxisaurus*, since the skull roof is generally flat in other thyreophorans (e.g., *Haubold, 1990*; *Sereno and Dong, 1992*; *Norman, 2020c*). However, given its uncertain identification, this element is not considered further herein.

### Braincase

The occipital portion of the skull is well preserved and its broadest part reaches a maximum width of approximately 134 mm (measured between the distal ends of the paroccipital processes). This is comparable to that of *Scelidosaurus* (NHMUK PV R1111: c. 120 mm; N.B. the scale bar given in *Norman, 2020c*, *Figure 3* is incorrect, implying that the holotype skull is twice as large as it is) and the Late Jurassic ankylosaurian *Gargoyleosaurus* (154 mm; *Carpenter et al., 1998*) but is substantially greater than the estimated total skull width of *Emausaurus* (83 mm; *Haubold, 1990*). In posterior view, the occipital bones appear to be completely fused with each other, and the junctions between them are obscured (*Figure 5D*). The dorsal half of the occiput is strongly inclined anteriorly. A robust nuchal crest immediately dorsal to the foramen magnum extends vertically to meet the parietal (*Figure 5D*) and is flatter and wider than that present in *Scelidosaurus* (NHMUK PV R1111; *Norman, 2020c*). A deep dorsoventrally elongated, subtriangular depression is present on each side of the nuchal crest, excavating the posterior surface of the supraoccipital (*Figure 5D*), likely representing insertion areas for the neck musculature. By contrast, the corresponding area in *Scelidosaurus* is very shallowly concave and coarsely textured (NHMUK PV R1111; *Norman, 2020c*). The same region bears only a shallow concavity in ankylosaurians (e.g., *Gargoyleosaurus*, *Pawpawsaurus*, and *Euoplocephalus*), and in stegosaurs this depression is shallow in *Huayangosaurus* and deep and subquadrate in *Stegosaurus* (*Gilmore, 1914*; *Sereno and Dong, 1992*; *Lee, 1996*; *Carpenter et al., 1998*; *Vickaryous and Russell, 2003*; *Norman, 2020c*). Consequently, these large, teardrop-shaped fossae are a potential autapomorphy of *Yuxisaurus*.

Dorsolateral to the foramen magnum, at the base of each paroccipital process, there is a broad fossa for the reception of the proatlas (*Figure 5D*). A pair of short, rough ridges diverge dorsolaterally

from the dorsal midline of the foramen magnum and separate the proatlantal fossae from the parasagittal depressions adjacent to the nuchal crest (*Figure 5D*). The paroccipital process of *Yuxisaurus* is strap-like, extending laterally and slightly posteriorly from each side of the foramen magnum, as in some ankylosaurians (such as *Pinacosaurus*: *Maryanska, 1971*) and stegosaurs (such as *Stegosaurus*: *Gilmore, 1914*), whereas in *Scelidosaurus* (NHMUK PV R1111; *Norman, 2020c*) and some ankylosaurians the paroccipital process extends ventrolaterally (*Vickaryous and Russell, 2003*). In *Yuxisaurus*, the ventral margin of the paroccipital process is straight on the left side but slightly concave on the right side (*Figure 5D*). The distal end of the process is dorsoventrally expanded but is asymmetrical, so that most of this expansion occurs dorsally rather than ventrally. This asymmetrical expansion creates a distinct, 'V'-shaped notch on the dorsal margin of the paroccipital process (*Figure 5D*). This notch appears to be unique to *Yuxisaurus* and is regarded as autapomorphic. By contrast, this margin is subtly concave in *Scelidosaurus* (NHMUK PV R1111; *Norman, 2020c*), convex in the early diverging ornithischian *Lesothosaurus* (*Sereno, 1991*) and is straight or slightly convex in stegosaurians and ankylosaurians (*Gilmore, 1914*; *Sereno and Dong, 1992*; *Vickaryous and Russell, 2003*; *Norman, 2020c*). On the left paroccipital process, at about the same level as the concavity, lies a tongue-like slit, resembling the condition in *Scelidosaurus*, where a spur-like process indicates the position of the posttemporal fenestra (NHMUK PV R1111: *Norman, 2020c*). However, this feature is absent on the right-hand side, which might be the result of taphonomic distortion. The paroccipital process contacts the squamosal anterodorsally and the quadrate anteroventrally but is not fused with them, similar to the condition in *Scelidosaurus* (NHMUK PV R1111; *Norman, 2020c*) and stegosaurs (*Gilmore, 1914*; *Sereno and Dong, 1992*), but differing from ankylosaurs like *Gargoyleosaurus*, *Talarurus*, *Pinacosaurus*, *Tarchia,* and *Euoplocephalus* in which these bones are fused (*Godefroit et al., 1999*; *Vickaryous and Russell, 2003*). In lateral view, the distal end of the paroccipital process is sinuous, with its thin ventral half curving posteriorly but the thick dorsal half anteriorly (*Figure 4A and B*).

The foramen magnum is subelliptical in outline, with its long axis extending horizontally. The aperture contains a rounded fragmentary bone, which probably represents the axial odontoid process. The occipital condyle was broken when separated from the cervical series, but its remaining portion suggests that it had a reniform outline, as also occurs in *Scelidosaurus* (NHMUK PV R1111; *Norman, 2020c*). Due to fusion, the relative contributions of the basioccipital and exoccipital to the boundaries of the foramen magnum cannot be determined.

In lateral view, the occipital condyle is set on a short neck and the ventral margin of the basioccipital curves anteroventrally (*Figure 4A*). Anterior to the occipital condyle, the ventral surface of the basioccipital is generally smooth but bears some irregular pits. The basioccipital expands laterally and especially ventrally to form prominent, rounded basal tubera, which are strongly offset ventrally with respect to the long axis of the occipital condyle (*Figure 4A*). This gives the posteroventral corner of the braincase a dorsoventrally deep, 'stepped' appearance in lateral view. By contrast, the basal tubera lie at the same level as, or slightly dorsal to, the occipital condyle in *Scelidosaurus* (NHMUK PV R1111; *Norman, 2020c*) and *Emausaurus* (*Haubold, 1990*). In ankylosaurs and stegosaurs, the basal tubera project only a short distance ventral to the occipital condyle (e.g., *Gilmore, 1914*; *Maryanska, 1977*; *Sereno and Dong, 1992*; *Vickaryous and Russell, 2003*) and it seems likely that the deep, 'stepped' basal tubera of *Yuxisaurus* are an autapomorphy. The basal tubera are widely separated in ventral view in *Yuxisaurus*, as also occurs in *Scelidosaurus* (NHMUK PV R1111; *Norman, 2020c*). However, the new taxon lacks the prominent midline ridge that lies between the basal tubera in *Scelidosaurus* (NHMUK PV R1111; *Norman, 2020c*). Dorsal to the basal tubera is a recess delineated by a sharp ridge anteriorly and another one posteriorly, which represents the otic region containing the fenestra ovalis and that is presumably formed by the otooccipital (*Figure 4A*) although bone boundaries in this region are impossible to assess due to fusion. Posterior to this recess, and bounded by the occipital condyle posteriorly, is another smaller recess, which is inferred to have contained the exits of cranial nerves IX–XI (the glossopharyngeal [IX], accessory [XI], and vagus nerves [X]). However, all of these inferred foramina are completely concealed by matrix and cannot be identified (*Figure 4A*).

The basisphenoid is preserved but is broken ventrally on its left-hand side. As in other thyreophorans, it is anteroposteriorly short in comparison with the basioccipital. Its base forms a gently curved shelf, posterolateral to which the anteroposteriorly compressed basipterygoid processes are directed ventrolaterally in posterior view and slightly posteriorly in lateral view (*Figures 4A and 5D*). The basipterygoid processes are situated considerably lower than the basal tubera in both lateral and

posterior views, creating an additional 'step' in the posterior margin of the braincase (*Figures 4A and 5D*). This differs from the conditions in *Scelidosaurus* (NHMUK PV R1111; *Norman, 2020c*), *Emausaurus* (*Haubold, 1990*), stegosaurs (e.g., *Gilmore, 1914*; *Galton, 1988*; *Sereno and Dong, 1992*), and ankylosaurs (e.g., *Maryanska, 1977*; *Vickaryous and Russell, 2003*), in which these processes only extend for a short distance ventrally with respect to the occipital condyle and are poorly exposed in posterior view, and this probably represents an additional autapomorphy of *Yuxisaurus*. Although the left basipterygoid process is missing, the processes appear to have been separated by an angle of 30° (*Figure 5D*), whereas this angle is closer to 60° in *Scelidosaurus* (NHMUK PV R1111; *Norman, 2020c*).

The basipterygoid and parasphenoid are fused indistinguishably and the cultriform process is lentiform in transverse cross-section. It protrudes anterodorsally for a short distance, but its anterior portion is broken (*Figure 4A*). As with other features of the basicranium, the cultriform process is ventrally offset with respect to the occipital condyle, contributing to the deep, stepped appearance of the braincase in lateral view (*Figure 4A*). In *Lesothosaurus* (NHMUK PV RU B17; *Porro et al., 2015*), *Scelidosaurus* (NHMUK PV R1111; *Norman, 2020c*), and *Huayangosaurus* (*Sereno and Dong, 1992*), the cultriform process and occipital condyle are in approximately the same plane.

The junction between the basisphenoid and prootic cannot be determined, but the presence of the latter can be inferred from the position of a large, teardrop-shaped foramen on the lateral surface of the braincase, which is inferred to be the exit for cranial nerve V (trigeminal: *Figure 4A*). Similarly, at least a portion of the laterosphenoid is present anterior to this opening, although no sutures are visible in this region. The braincase is open anteriorly, revealing the endocranial cavity, which is vertically expanded and has a rounded, smooth inner surface (*Figure 5C*). A bone fragment attached to the anterior border of the right laterosphenoid is identified as the right orbitosphenoid. Ossified orbitosphenoids are also present in *Scelidosaurus* (NHMUK PV R1111; *Norman, 2020c*), ankylosaurs (*Maryanska, 1977*; *Vickaryous and Russell, 2003*), and stegosaurs (*Gilmore, 1914*).

## Mandible

The post-dentary portions of both hemimandibles are preserved, including the angulars, surangulars, prearticulars, and articulars (*Figure 7*). Their lateral surfaces are smooth and bear no ornamentation or fused osteoderms.

In lateral view, the ventral margin of the angular is very slightly concave, but its posterior part curves posterodorsally at an angle of approximately 155°, as in *Scelidosaurus* (*Norman, 2020c*). The angular is tallest anteriorly but tapers posteriorly and has an almost straight dorsal margin that turns abruptly dorsally close to its posterior end (*Figure 7A*). The elongated, upturned posterior process of the angular is not present in either *Emausaurus* (*Haubold, 1990*) or *Scelidosaurus* (NHMUK PV R1111; *Norman, 2020c*) and appears to be unique to *Yuxisaurus* among early branching thyreophorans; it is regarded as a potential autapomorphy herein. Viewed ventrally, the angular of *Yuxisaurus* has a tapering posterior terminus (*Figure 7B*), and the sinuous suture with the prearticular extends along the ventral margin, which can only be seen beneath the adductor fossa in medial view (*Figure 7C and G*). The smooth lateral surface bulges laterally at its center, which is more prominent on the right side, but the ventral surface is generally flat.

In lateral view, the surangular has subparallel dorsal and ventral margins. Both margins are horizontal and straight posteriorly, but curve anterodorsally anteriorly (*Figure 7A*). Along the dorsal border immediately anterior to this inflexion is a dorsally extending process, with a sharp dorsal margin that also bulges slightly laterally. On the left surangular, the anterior portion of this process curves medially while the posterior portion is missing. By contrast, this process is oddly shaped on the right side, having a broad, subtriangular base with a transversely wide but anteroposteriorly compressed process that is posterodorsally directed (*Figure 7E and H*). Further anteriorly, the dorsal margin of the surangular expands transversely, to roof the adductor fossa medially, and laterally to overhang the lateral surface. In lateral view, this dorsal expansion extends anterodorsally, whereas it is generally horizontal in *Emausaurus* and *Scelidosaurus* (*Haubold, 1990*; *Norman, 2020c*). The surface ventral to the lateral overhang is broadly depressed, and its posterodorsal corner is pierced by a foramen (*Figure 7E*). This foramen is prominent on the right hemimandible but cannot be identified on the left side. Further anteriorly, the surangular dorsal margin forms a dorsal apex. Its medial margin is higher than its lateral margin in dorsal view, so that its dorsal surface is oriented laterally. This apex, presumably the highest point of the mandible, flattens anterolaterally and the dorsal surface anterior to this apex is generally flat. Immediately beneath this apex the lateral surface bulges strongly laterally

(*Figure 7E*). In medial view, the surangular encloses the ovoid adductor fossa dorsally and posteriorly (*Figure 7G*). The inner surface of the adductor fossa is smooth but it bears an irregular vertical ridge in the center of its ventral half (*Figure 7G*). As with the articular surface of the hemimandible, the surangular curves medially posteriorly and expands medially to form an elevated platform relative to the articular surface, and then shrinks abruptly, tapering posteromedially (*Figure 7D and H*). At the inflection point of this process the lateral surface bulges laterally, posterior to which the lateral surface bears an anterolaterally-posteromedially elongated depression that is prominent on both hemimandibles (*Figure 7A and E*).

In medial view, the prearticular forms the ventral margin of the adductor fossa. Its dorsal margin is concave and sharp, but is interrupted by a rounded process that lies slightly posterior to the middle of the fossa, as also occurs in *Scelidosaurus* (*Norman, 2020c*). The prearticular presumably contributed to the posterior margin of the adductor fossa, but the extent of this cannot be recognized in this specimen. Adjacent to the posterior margin, the prearticular bears a dorsal concavity, which expands laterally to form a broad, flattened articular surface that meets the surangular laterally (*Figure 7C and G*). Sutures are difficult to determine in this region but it seems likely that the prearticular extended posteriorly to the end of the mandible and completely fused with the surangular ventrally.

The articular is completely fused with the surrounding bones so its original outline is unknown. Nevertheless, in dorsal view, the articular bears a concavity medially, which is broader on the right hemimandible than on the left (*Figure 7D and H*). Anterior to this concavity, the articular has a dorsal pyramidal process. Posteriorly the articulars have different shapes on different sides, as the right articular possesses a mediodorsal flange with a flat dorsal surface, while the left articular has a vertical flange and bears a deep fossa on the dorsal surface (*Figure 7D and H*).

A bone fragment in the anterodorsal corner of the adductor fossa of the right hemimandible might represent part of a coronoid, but further information is unavailable due to poor preservation.

## Dentition

Most of the maxillary tooth crowns were abraded away accidentally during preparation (*Figure 3E*). The alveolar sockets are elliptical and slightly expanded transversely (*Figure 3D*). Most of the teeth are similar in size except for the 7th, 8th, 10th, and 11th teeth, which appear to be slightly larger on the basis of their cross-sections (approximately 5-mm labiolingually by 7-mm mesiodistally). The best-preserved tooth is the posterior-most one, which is embedded in its socket. This tooth crown is triangular in lingual view and has coarsely denticulate mesial and distal margins (*Figure 3F*). Its lingual surface is ornamented with multiple (at least four) pairs of vertical ridges lying in parallel to each another, which extend to the ventral margin of the crown and support the marginal denticles. These ridges are narrow but densely packed and are almost evenly distributed over the crown surface. The tooth differs from those of *Lesothosaurus* and *Scutellosaurus*, which lack ridges on the crown surface (*Colbert, 1981*; *Sereno, 1991*; *Breeden et al., 2021*), and those of *Emausaurus* and *Scelidosaurus*, which have only incipient fluting and ridges (*Haubold, 1990*; *Norman, 2020c*; NHMUK PV R1111). However, the teeth of many ankylosaurs (*Vickaryous and Russell, 2003*) and stegosaurs (*Galton, 2004*) do bear numerous ridges, although *Yuxisaurus* lacks the prominent primary ridge that is often present in stegosaurs as well as the rounded denticles usually present in the latter clade.

## Axial skeleton

An articulated series of the four anterior-most cervical vertebrae is present and well preserved (*Figure 8*). Originally, these vertebrae were articulated with the occiput, but they were separated during preparation. Five isolated dorsal vertebrae of varying preservation are also present (*Figure 9*). They were not articulated and their exact sequence cannot be confirmed due to the variation in vertebral morphology and proportions that occurs along the dorsal series of other thyreophoran dinosaurs. However, we attempt to place them in relative order herein.

### Cervical vertebrae and ribs

The atlas is composed of a ventral intercentrum and a pair of dorsal neural arches (*Figure 8A–D, F and G*). In anterior view, the atlas is rotated clockwise through 30° with respect to the other preserved vertebrae (*Figure 8F*). It is much wider transversely than long anteroposteriorly . The intercentrum is crescentic to reniform in outline in anterior view and possesses an anterior articular surface that is

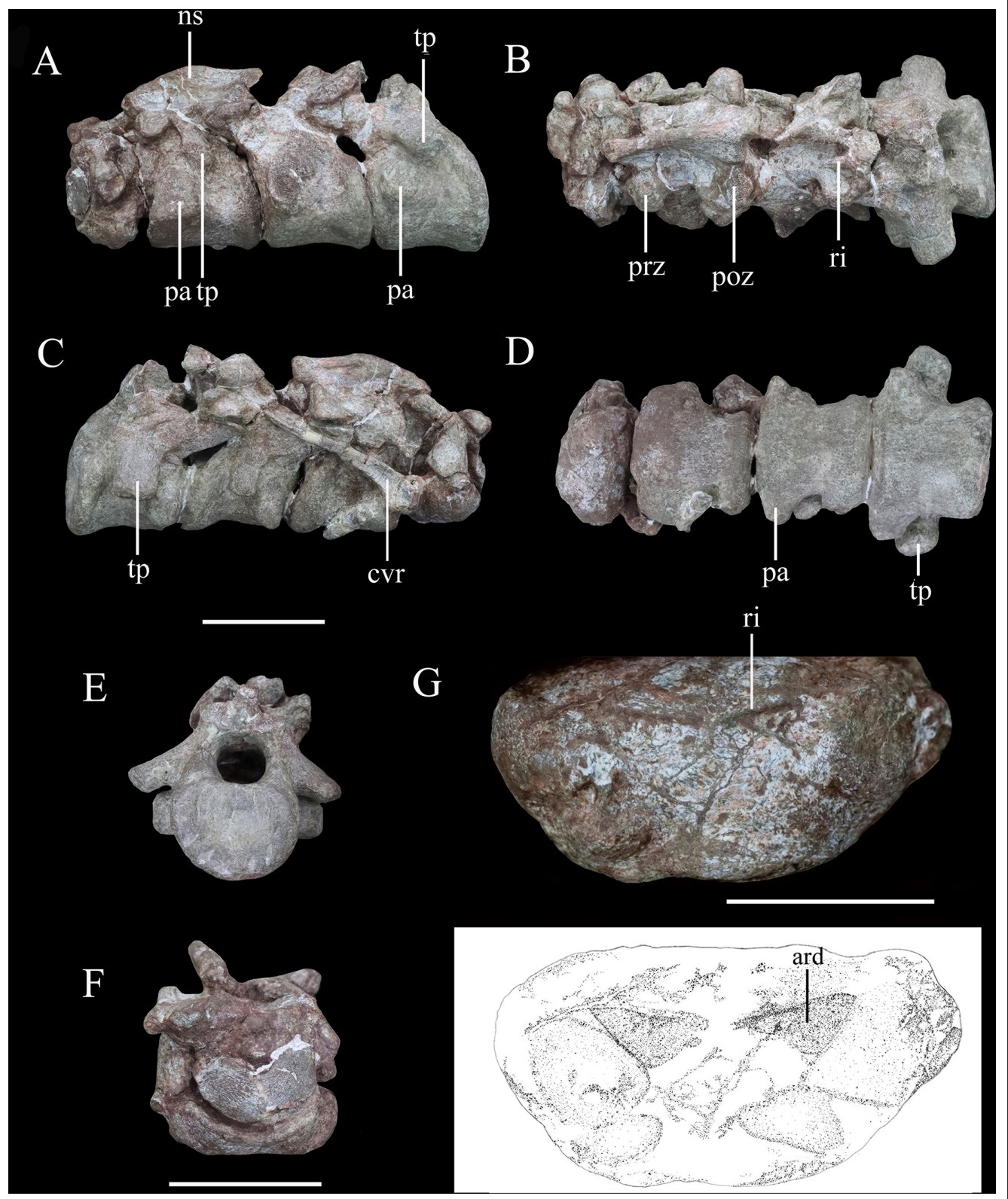

**Figure 8.** Articulated series of the anterior-most cervical vertebrae (atlas, axis, and cervicals 3 and 4) of *Yuxisaurus kopchicki* in (**A**) left lateral, (**B**) dorsal, (**C**) right lateral, (**D**) ventral, (**E**) posterior, and (**F**) anterior views. Atlas in (**G**) ventral view with interpretative diagram beneath. Abbreviations: ard, arrow-like depression; cvr, cervical rib; ns, neural spine; pap, parapophysis; poz, postzygapophysis; prz, prezygapophysis; ri, ridge; tp, transverse process. Scale bar equals 5 cm.

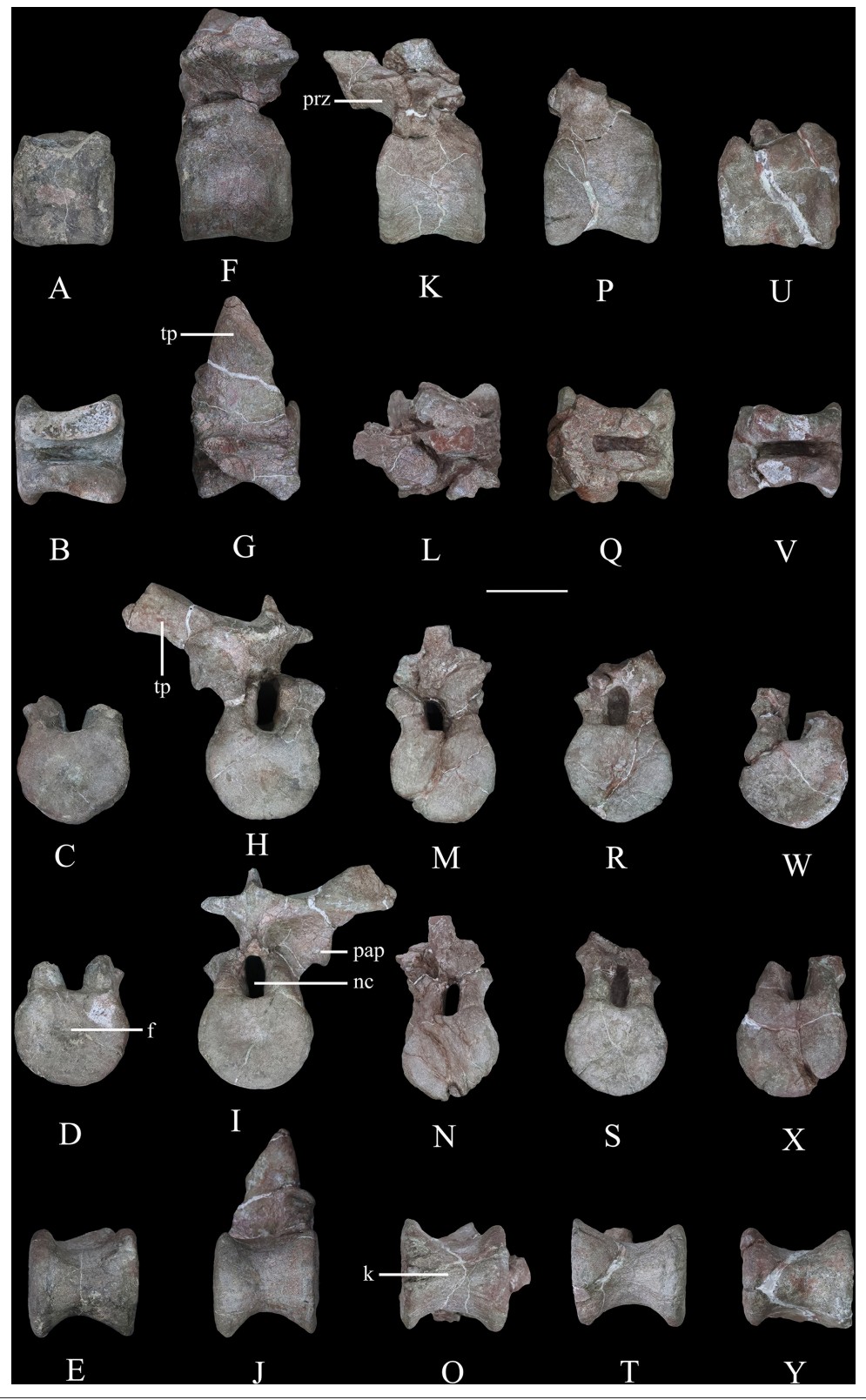

**Figure 9.** Dorsal vertebrae of *Yuxisaurus kopchicki*. D1 in (**A**) left lateral, (**B**) dorsal, (**C**) anterior, (**D**) posterior, and (**E**) ventral views. D2 in (**F**) left lateral, (**G**) dorsal, (**H**) anterior, (**I**) posterior, and (**J**) ventral views. D3 in (**K**) left lateral, (**L**) dorsal, (**M**) anterior, (**N**) posterior, and (**O**) ventral views. D4 in (**P**) left lateral, (**Q**) dorsal, (**R**) anterior, (**S**) posterior, and (**T**) ventral views. D5 in (**U**) left lateral, (**V**) dorsal, (**W**) anterior, (**X**) posterior, and (**Y**) ventral views. Abbreviations:

*Figure 9 continued on next page*

*Figure 9 continued*

f, fossa; k, keel; nc, neural canal; pap, parapophysis; prz, prezygapophysis; tp, transverse process. Scale bar equals 5 cm.

broadly concave and faces anterodorsally (*Figure 8F*). A massive but short swelling is present on either side of the lateral surface that is directed ventrally and laterally, anterior to which is a low anterodorsally directed ridge (*Figure 8F and G*). This ridge is separated from the swelling by a distinct anterodorsally directed trough. Viewed ventrally, a pair of arrow-like depressions, which point posterolaterally, occupies the posterior-most surface of the intercentrum to form sharp posterior margins (*Figure 8G*). This feature appears to be unique to *Yuxisaurus* and is absent in *Scelidosaurus* (NHMUK PV R1111; *Norman, 2020a*) and *Scutellosaurus* (*Breeden and Rowe, 2020*; *Breeden et al., 2021*). In contrast to *Yuxisaurus*, the ventral surface of the atlantal intercentrum in *Stegosaurus* bears two posterolaterally directed ridges and a subtle midline ridge separating two cavities (*Maidment et al., 2015*). The left neural arch is incompletely preserved but resembles closely the right one where preserved. The right pedicle is cylindrical, with an expanded ventral base articulating with the intercentrum. The postzygapophysis is a thin plate, extending posterodorsally, as in *Scelidosaurus* (NHMUK PV R1111; *Norman, 2020a*) and *Gastonia* (*Kinneer et al., 2016*), but its lateral margin bulges and thickens. A small plate above the neural arch probably represents the proatlas.

The axial centrum is massive and approximately equally long and wide (*Figure 8A, C and D*). Viewed laterally both its anterior and posterior articular surfaces are inclined anteriorly, giving it a trapezoidal outline (*Figure 8A and C*). Its anterior articular surface is strongly concave but the posterior surface appears to be flatter. The anteroventrally placed triangular parapophysis is prominent, expanding laterally, posterior to which a distinct depression extends over the lateral surface (*Figure 8A*). The ventral surface of the centrum is smooth, with a rounded ridge in the center that is flanked by oblique surfaces laterally (*Figure 8D*), similar to the condition in *Scutellosaurus* (*Breeden et al., 2021*). By contrast, the axial centra of *Scelidosaurus* (NHMUK PV R1111; *Norman, 2020a*), *Stegosaurus* (NHMUK PV R36730; *Maidment et al., 2015*), and *Gargoyleosaurus*, all bear a midline keel. In lateral view, the left diapophysis is directed ventrally (*Figure 8A*), but its tip is separated from the parapophysis by an anterodorsally extending trough. The right diapophysis is concealed by a cervical rib and surrounding matrix. In other thyreophorans, such as *Scelidosaurus* (NHMUK PV R1111; *Norman, 2020a*), *Stegosaurus* (NHMUK PV R36730; *Maidment et al., 2015*), and *Sauropelta* (*Vickaryous and Russell, 2003*), the diapophysis is directed ventrally but also slightly laterally and can be seen in ventral view. Both of the prezygapophyses curve laterally and ventrally and bear slightly convex articular facets (*Figure 8A and B*). Due to rotation of the atlas (see above), the right prezygapophysis of the axis does not articulate with the corresponding atlantal postzygapophysis. The postzygapophysis expands and diverges laterally to a greater degree than the prezygapophysis in dorsal view (*Figure 8B*). Its articular facet faces ventrally and is slightly concave as in the ankylosaur *Sauropelta* (*Vickaryous and Russell, 2003*), but differs from *Scelidosaurus* in which the articular facet faces ventrolaterally (NHMUK PV R1111; *Norman, 2020a*). A flat lamina above the diapophysis connects the base of the prezygapophysis anteriorly and the postzygapophysis posteriorly. The thick neural spine extends anteroposteriorly with a mild anterior transverse expansion but flares posteriorly where the postzygapophysis meets the spine on either side. In *Scelidosaurus* (NHMUK PV R1111; *Norman, 2020a*), the anterior transverse expansion is much more prominent than in *Yuxisaurus*, whereas in the ankylosaur *Sauropelta* (*Vickaryous and Russell, 2003*) and the stegosaur *Stegosaurus* (NHMUK PV R36730; *Maidment et al., 2015*), this expansion appears to be mild. In lateral view, the dorsal margin of the axial neural spine is sinusoidal with a central apex, an anterior portion that slopes ventrally and that is nearly straight, and a posterior portion that is slightly concave (*Figure 8A and C*), similar to that of *Sauropelta* (*Vickaryous and Russell, 2003*) and *Stegosaurus* (NHMUK PV R36730; *Maidment et al., 2015*). In contrast, the dorsal margin of the axial neural spine is convex in *Lesothosaurus* (NHMUK PV R11004; *Baron et al., 2017b*) and straight in *Scelidosaurus* (NHMUK PV R1111; *Norman, 2020a*) and *Scutellosaurus* (*Breeden et al., 2021*). Both the anterior and posterior ends of the neural spine overhang the articular surfaces slightly in lateral view, as seen also in *Lesothosaurus* (NHMUK PV R11004; *Baron et al., 2017b*) and some ankylosaurs (*Vickaryous and Russell, 2003*). By contrast, in *Scelidosaurus* the posterior end of the neural spine extends much farther than the posterior

articular surface (NHMUK PV R1111; *Norman, 2020a*). Posteriorly, a deep, oval postspinal fossa is present, as also occurs in *Scelidosaurus* and *Stegosaurus* (NHMUK PV R1111, NHMUK PV R36730; *Maidment et al., 2015*; *Norman, 2020a*).

The third cervical vertebra is similar in size to the axis. The centrum is spool-shaped and constricted in the middle (*Figure 8D*). Its ventral surface possesses a rounded ridge that extends anteroposteriorly, contrasting with the presence of a keel in *Scelidosaurus* (*Norman, 2020a*) and *Scutellosaurus* (*Breeden and Rowe, 2020*; *Breeden et al., 2021*). In lateral view, the centrum is relatively short and subquadrate in outline (*Figure 8A and C*), with a length to posterior height ratio of approximately 1.4, similar to the condition in some ankylosaurs (*Maleev, 1956*; *Kilbourne and Carpenter, 2005*), but contrasting with the more elongate cervicals present in *Scelidosaurus* (~1.7; *Norman, 2020a*), *Scutellosaurus* (~2.1; *Breeden and Rowe, 2020*), and some stegosaurs (NHMUK PV R36730; *Maidment et al., 2015*). The parapophysis is not as prominent as that on the axis, and is a rounded process occupying the anterior corner of the lateral surface, posterior to which the lateral surface of the centrum is depressed. The right diapophysis curves ventrolaterally and its distal end is crescentic with a flat dorsal surface and a convex ventral margin, as in *Scelidosaurus* (*Norman, 2020a*). The prezygapophyses extend anterodorsally beyond the centrum anterior margin. The postzygapophyses project posterodorsally and somewhat laterally, terminating flush with the posterior margin of the centrum, and their articular facets face ventrolaterally. The dorsal surface of the postzygapophysis bears a rugose ridge that expands transversely as it extends posteriorly, as also occurs in *Scelidosaurus* (*Norman, 2020a*). The neural spine is damaged, but it appears to have expanded strongly posteriorly to overhang the posterior margin of the centrum (*Figure 8A–C*). This feature is absent in *Scelidosaurus* (*Norman, 2020a*), in which the neural spine terminates more anteriorly, but is present in some cervicals of *Scutellosaurus* (*Breeden and Rowe, 2020*). A postspinal fossa is present but is smaller than that of the axis.

The fourth cervical centrum is similar to that of the preceding vertebra, both in overall morphology and proportions (*Figure 8A–D*). The lateral surface posterior to the parapophysis bears the shallowest excavation of the four preserved cervicals. The posterior articular surface has a crescentic outline, with a flat upper margin and ventral convex margin, and its center is occupied by a semilunate concavity (*Figure 8E*). The parapophysis is cylindrical in outline, differing from those of the axis and third cervical, which have subtriangular and rounded outlines, respectively. The transverse process extends ventrolaterally and has an elliptical cross-section (*Figure 8A–D*). The prezygapophysis projects anterodorsally to a point almost halfway along the preceding cervical centrum (*Figure 8A*), contrasting with the shorter processes present in *Scelidosaurus* (*Norman, 2020a*), *Scutellosaurus* (*Breeden and Rowe, 2020*; *Breeden et al., 2021*), and *Stegosaurus* (*Maidment et al., 2015*), but it is unclear if this has been altered taphonomically. A postspinal fossa is present, but is the smallest found in the preserved cervicals. The large neural canal is rounded in outline (*Figure 8E*).

A cervical rib articulates with the parapophysis of the right axis and, partly, with the posterior surface of the atlas via its expanded single head. Its elongate shaft extends posterodorsally at an angle of 32° from the horizontal with a gentle curvature (*Figure 8C*), almost reaching the middle of the third cervical with a total length of about 75 mm. By contrast, the axial ribs of *Scelidosaurus* are relatively shorter (*Norman, 2020a*), but they are unknown in other early thyreophoran taxa. In *Yuxisaurus*, the rib shaft is transversely compressed and tapers distally, but that of *Scelidosaurus* is more rod-like (*Norman, 2020a*) but this difference could reflect taphonomic compression. The lateral surface of the rib shaft is generally flat, but is slightly depressed anteriorly, and is separated from the head by a shallow break-in-slope in lateral view. By contrast, the axial rib of *Scelidosaurus* bears a lateral ridge along the shaft (*Norman, 2020a*). Another 26-mm-long rib fragment is attached to the lateroventral surface of the axis.

## Dorsal vertebrae

Five isolated dorsal vertebrae of varying preservation are present (*Figure 9*) and are labeled as D1–D5 for convenience. They are generally similar to those of a range of thyreophoran taxa, including *Scelidosaurus* and ankylosaurs (*Vickaryous and Russell, 2003*; *Norman, 2020a*), although they lack the extreme neural arch elongation of stegosaurs (*Galton, 2004*).

'D1' preserves the centrum and the bases of the neural arch pedicles only (*Figure 9A–E*). Its anterior articular surface is concave (*Figure 9C*) while the posterior articular surface is flat but possesses a rounded fossa in the center (*Figure 9D*). Both articular surfaces are subcircular in outline. The centrum

has a subquadrate outline in lateral view (*Figure 9A*), is spool-shaped in ventral view, and lacks a ventral keel (*Figure 9E*).

The centrum of 'D2' is spool-shaped with a ventral margin that is gently arched in lateral view (*Figure 9F–J*). The ventral surface is rounded and lacks a keel (*Figure 9J*). Both articular surfaces are subcircular in outline but with a slightly flattened dorsal margin (*Figure 9H1*). The anterior articular surface appears to be more dorsoventrally compressed than the posterior one. The anterior articular surface is concave, while the posterior surface is nearly flat with its center occupied by a distinct concavity. A partial neural arch is present. The parapophysis is positioned level with the dorsal part of the neural canal and is an expanded oval facet that is situated close to the anterior rim of the centrum in lateral view (*Figure 9F*). Its diapophysis is stout and projects laterodorsally at an angle of ~33° above the horizontal. Its dorsal surface is generally flat with a gentle swelling in the middle. Although broken, the neural spine appears to have been low, with a transverse expansion anteriorly, and is nearly level with the diapophysis in height in lateral view (*Figure 9F*). The neural canal is ovoid in outline and dorsoventrally elongated (*Figure 9H1*). All of the zygapophyses are missing, but a broad infrapostzygapophyseal fossa is present (*Figure 9F*).

'D3' has an amphicoelous, spool-shaped centrum (*Figure 9K–O*). In lateral view its ventral margin is more arched than that of 'D2' (*Figure 9K*), and its ventral surface is constricted into a keel (*Figure 9O*). The left lateral surface bears an anteroposteriorly elongated depression on its dorsal part, but this is absent on the right-hand side. A partial neural arch is present. The remaining portion of the left diapophysis is horizontally inclined and has a flat dorsal surface. The neural spine is thickened medio-laterally, with a transverse width of 16 mm in the middle, which is significantly greater than that of 'D2' (4 mm). The thickened neural spine and horizontal transverse process suggest that this is most likely a posterior dorsal vertebra (*Norman, 2020a*). The prezygapophysis curves anterodorsally from the base of the neural spine, overhanging the anterior margin of the centrum (*Figure 9K*). Its articular facet was probably directed dorsally but is concealed by an adhered fragment of the preceding postzygapophysis. In anterior view, the infraprezygapophyseal surface is broadly concave (*Figure 9M*). The postzygapophyseal fragment of the preceding vertebra is massive, extending across the vertebral midline, suggesting that the postzygapophysis fused with its counterpart in the posterior dorsal series.

'D4' consists of a centrum and partial neural arch lacking processes (*Figure 9P–T*). The centrum is slightly longer than that of 'D3', but its morphology is generally similar, including the presence of a ventral keel (*Figure 9T*). Its right lateral surface bears a shallow, elongate depression, but this is absent on the left. The remnant of the left parapophysis indicates that it was positioned high on the neural arch, immediately above the neural canal. Viewed anteriorly both of the neural arch pedicles are stout and have lateral margins that curve dorsally and then laterally as also occurs in 'D2' and 'D3' (*Figure 9R*).

'D5' consists only of the centrum and the broken bases of the neural arch pedicles (*Figure 9U–Y*). It is generally similar to the other dorsal vertebrae and is of equal length to 'D4,' although its concave lateral surfaces are smooth and lack depressions. The ventral margin of the centrum is only slightly concave in lateral view, and in ventral view, the keel is less prominent than that of 'D4' (*Figure 9Y*). The presence of ventral keels in posterior dorsal vertebrae contrasts with their absence in *Scelidosaurus* (*Norman, 2020a*) and *Stegosaurus* (*Maidment et al., 2015*), although some ankylosaurs have keeled posterior dorsal centra (*Kirkland and Carpenter, 1994*; *Kirkland et al., 2013*).

## Appendicular skeleton

The specimen includes limited appendicular elements, including: the proximal part of a left scapula and the distal part of the right scapula (*Figure 10*); a complete right humerus (*Figure 11*); and the distal part of the left femur (*Figure 12*).

### Scapula

The scapula is represented by a right scapula blade (*Figure 10A–D*) and a left proximal plate (*Figure 10E–H*), but unfortunately, these two pieces do not overlap in morphology so the overall shape and size of the scapula remain unclear. However, on the basis of the preserved parts, we estimate that a complete scapula would have been at least 475 mm long.

The left proximal plate of the scapula is poorly preserved with broken margins (*Figure 10E–H*). As preserved, it has a maximum width of ~188 mm. It is expanded dorsoventrally with respect to the

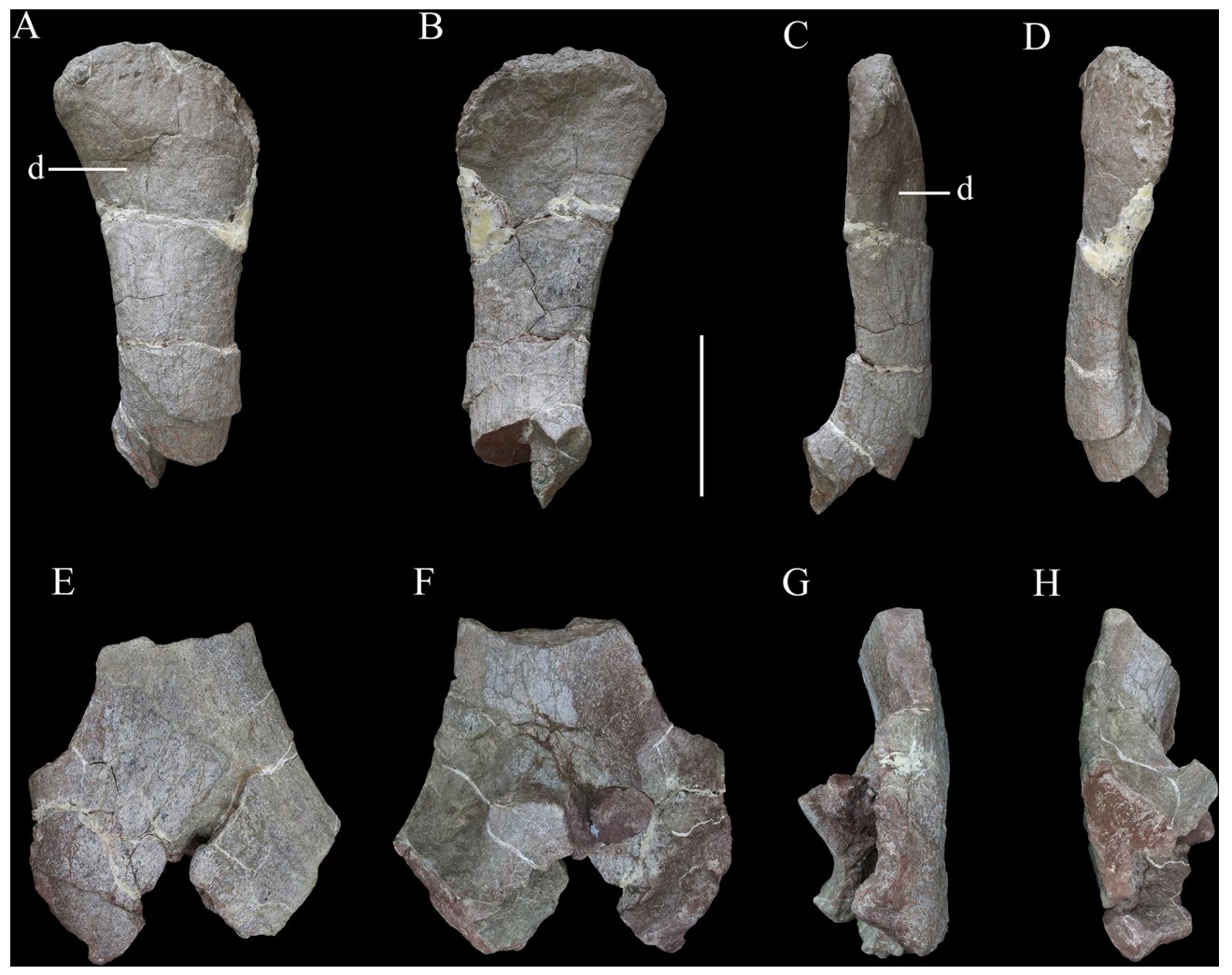

**Figure 10.** Scapulae of *Yuxisaurus kopchicki*. Distal part of right scapula in (**A**) lateral, (**B**) medial, (**C**) ventral, and (**D**) dorsal views. Proximal part of left scapula in (**E**) lateral, (**F**) medial, (**G**) ventral, and (**H**) dorsal views. Abbreviation: d, depression. Scale bar equals 10 cm.

scapula shaft and its lateral surface is shallowly convex. Anteriorly, a portion of the glenoid fossa is present, which is anteroposteriorly concave. An anteroposteriorly elongated depression occupies the ventral surface immediately posterior to the glenoid on the medial surface of the proximal end, as also occurs in *Gastonia* (**Kinneer et al., 2016**). The medial surface of the proximal scapula is strongly convex (*Figure 10F*). Few other details are available due to damage.

The scapula blade is relatively thick transversely, with a convex lateral surface and a flat or slightly depressed medial surface. In lateral view, its distal end is expanded dorsoventrally, with a maximum distal width of ~138 mm and a mid-shaft width of ~83 mm (*Figure 10A–D*). The dorsal and ventral margins of the scapula blade are subparallel along most of its length in lateral view, but the dorsal margin diverges slightly to contribute to the distal expansion, while the ventral margin curves ventrally at its distal end, so that the distal expansion is slightly asymmetrical with respect to the scapula long-axis. The distal margin is gently convex. This produces a scapula blade outline similar to those of *Scutellosaurus* (**Breeden and Rowe, 2020**), *Scelidosaurus* (**Norman, 2020a**), and some stegosaurs (**Galton, 2004**), while in most ankylosaurs, such as *Gargoyleosaurus*, *Sauropelta,* and *Gastonia*, the dorsal scapular margin almost parallels, the ventral margin and curves posteroventrally (**Godefroit et al., 1999**; **Vickaryous and Russell, 2003**; **Kinneer et al., 2016**), and in *Stegosaurus* these margins are essentially subparallel along their entire lengths (**Maidment et al., 2015**). Close to the distal end, the lateral surface bears a broad depression, but it is not clear if this is an original feature or due to

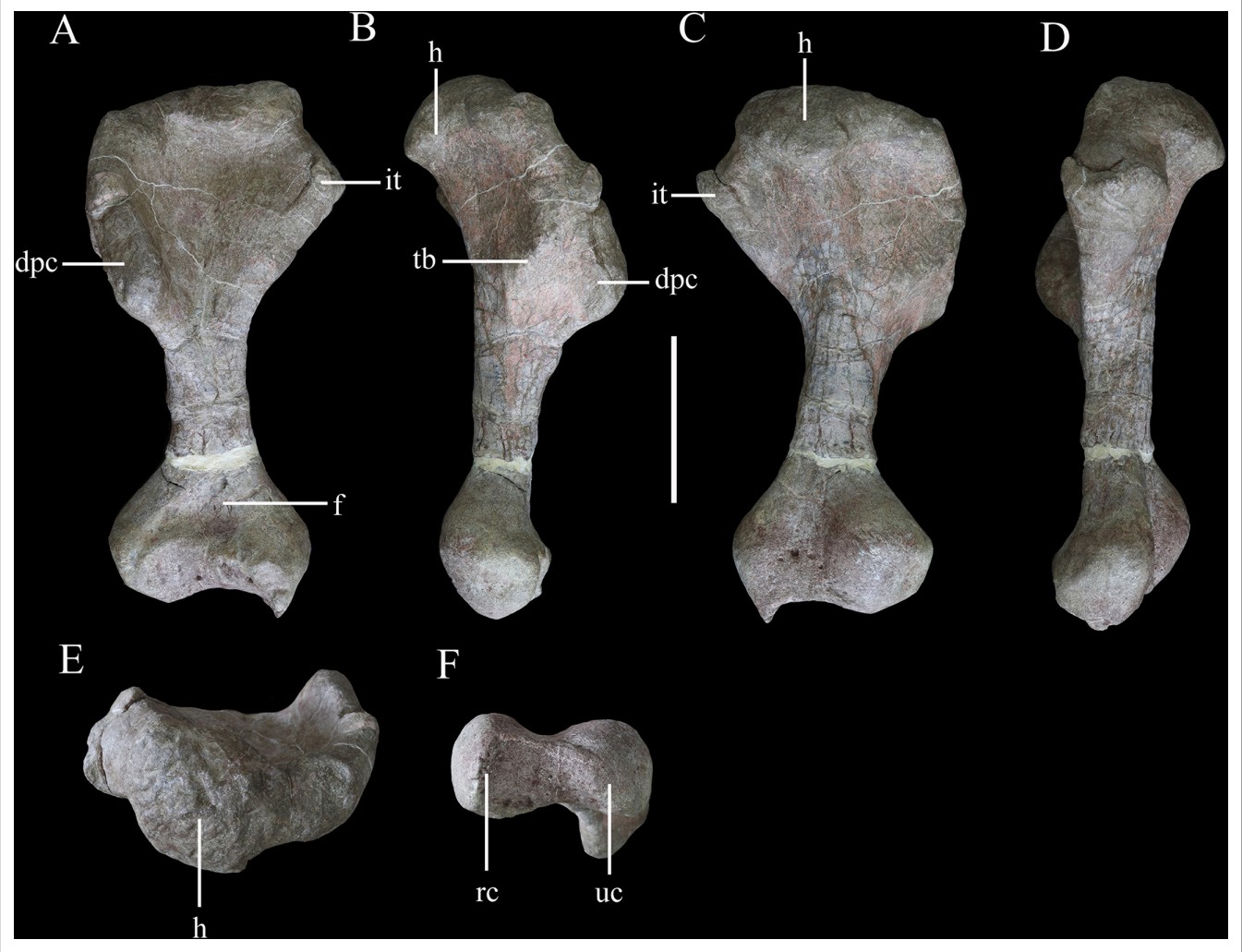

**Figure 11.** Right humerus of *Yuxisaurus kopchicki* in (**A**) anterior, (**B**) lateral, (**C**) posterior, (**D**) medial, (**E**) proximal, and (**F**) distal views. Abbreviations: dpc, deltopectoral crest; f, fossa; h, humeral head; it, internal tuberosity; rc, radial condyle; tb, tubercle; uc, ulnar condyle. Scale bar equals 10 cm.

taphonomic damage as there is some cracking in the area (*Figure 10A*). In dorsal or ventral views, the scapula blade is bowed, with the distal end inclined medially (*Figure 10C and D*).

### Humerus

The right humerus is well preserved, except for a small section of the distal end (*Figure 11*). It has an elongate, slender shaft, with a diameter of ~50 mm, which separates the proximal and distal expansions, which reach maximum widths of ~160 mm and 120 mm, respectively (*Figure 11A and C*). Both of these expansions are relatively broader than in either *Scutellosaurus* (*Colbert, 1981*; *Breeden and Rowe, 2020*; *Breeden et al., 2021*) or *Scelidosaurus* (*Norman, 2020a*), giving the humerus of *Yuxisaurus* a stockier appearance that is much more similar to those of ankylosaurs and stegosaurs (*Vickaryous and Russell, 2003*; *Galton, 2004*).

In anterior view, the humerus is straight, with the shaft lacking any significant deflection, and has a total length of ~345 mm (*Figure 11A*). A robust deltopectoral crest arises from the lateral margin of the proximal expansion and curves anteriorly and slightly medially, terminating in a thickened, transversely expanded distal end (35 mm in thickness). The deltopectoral crest extends to a point ~46% of the humeral length (*Figure 11A*). This is similar to the conditions present in *Scelidosaurus* (*Norman, 2020a*) and some ankylosaurs (e.g., *Pawpawsaurus*, *Europelta*: *Lee, 1996*; *Kirkland et al., 2013*), but differs from those of other ankylosaurs (e.g., *Pinacosaurus*, *Saichania*), where this crest terminates more distally (*Maryanska, 1977*; *Godefroit et al., 1999*), and *Scutellosaurus*, where it ends more

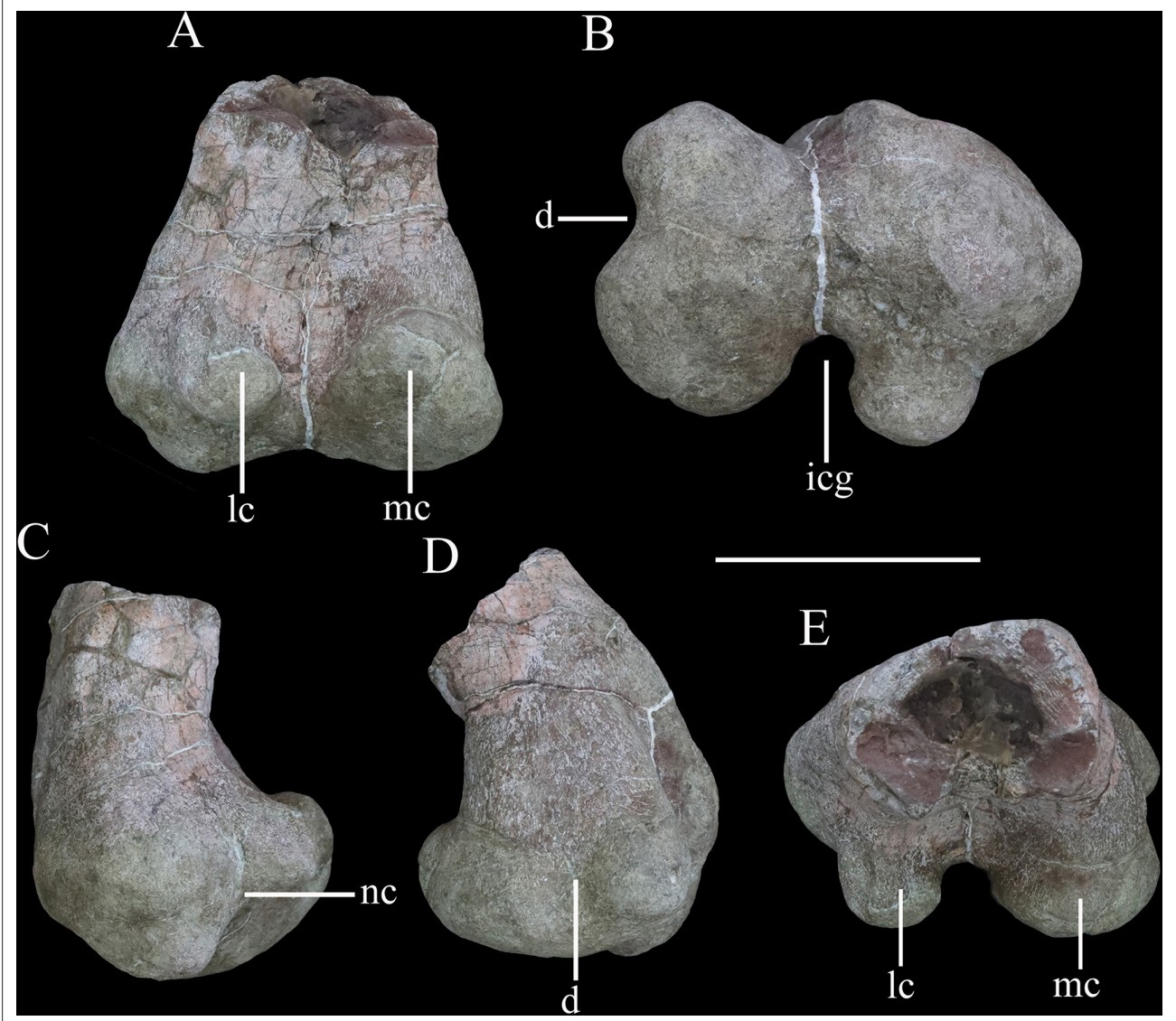

**Figure 12.** Distal end of right femur of *Yuxisaurus kopchicki* in (**A**) posterior, (**B**) ventral, (**C**) lateral, (**D**) medial, and (**E**) dorsal views. Abbreviations: d, depression; icg, intercondylar groove; lc, lateral condyle; mc, medial condyle; nc, notch. Scale bar equals 10 cm.

proximally (*Breeden et al., 2021*). The proximal anterior surface is strongly concave. In anterior view, the internal tuberosity has a straight, steeply inclined dorsomedial margin, which meets its curved ventromedial margin at an angle of ~110°. In proximal view, the internal tuberosity is anteroposteriorly expanded and is separated from the humeral head by a distinct notch dorsally (*Figure 11E*). This notch is absent in *Scelidosaurus* (*Norman, 2020a*) and *Scutellosaurus* (*Breeden and Rowe, 2020*; *Breeden et al., 2021*), but is present in some ankylosaurians (*Vickaryous and Russell, 2003*). The humeral head is subspherical, protrudes posteriorly and somewhat anteriorly with respect to the rest of the proximal end, and its posterior end curves posterolaterally, forming a triangular process (*Figure 11E*). This process partially encloses a posterolateral concavity, which is present in *Scelidosaurus* (*Norman, 2020a*) and *Europelta* (*Kirkland et al., 2013*) but not *Stegosaurus* (*Maidment et al., 2015*) or *Scutellosaurus* (*Breeden et al., 2021*). In posterior view, the proximal surface is convex, and a broad swelling arises from the base of the humeral head that extends ventrally for a short distance (*Figure 11C*). On the posterior surface of the deltopectoral crest, there is a large triceps tubercle, which is obliquely oriented and has a sharp, pointed apex (*Figure 11B*). This tubercle is present in various ankylosaurs, such as *Gastonia* and *Gargoyleosaurus*, and is supposedly homologous with a pocket-like muscle scar

present in *Scelidosaurus* (*Norman, 2020a*), but is absent in *Lesothosaurus* (*Baron et al., 2017b*) and *Scutellosaurus* (*Colbert, 1981*; *Breeden et al., 2021*).

The shaft has a subtriangular cross-section in its mid-part, with a flat anterior surface and convex posterior surface. Distally, the medial (ulnar) condyle extends further ventrally than the lateral condyle and also exhibits greater anteroposterior expansion. A broad, shallow, 'U'-shaped fossa is positioned immediately dorsal to the distal condyles on the anterior surface (*Figure 11A*), which differs from the longer, narrower, 'V'-shaped and shallower fossa seen in other early thyreophorans (*Breeden and Rowe, 2020*; *Norman, 2020a*; *Breeden et al., 2021*) and stegosaurs (*Maidment et al., 2015*), although a similar fossa occurs in some ankylosaurs (*Vickaryous and Russell, 2003*). A narrow, vertical depression separates the two condyles on the posterior surface (*Figure 11C*). In ventral view, the distal end has a dumbbell-shaped outline, though the ulnar condyle is more strongly expanded anteroposteriorly than the radial condyle (*Figure 11F*).

## Femur

The distal end of the left femur is preserved (*Figure 12*). It reaches a maximum transverse width of ~151 mm and is ~110 mm in anteroposterior length. The distal end is mediolaterally and anteroposteriorly expanded with respect to the preserved part of the femoral shaft (*Figure 12A*). The shaft has a subrectangular cross-section (*Figure 12E*). The anterior surface of the distal femur is generally flat, but its medial part is damaged. The distal end is divided into two articular condyles (*Figure 12B*). In posterior view, the lateral condyle is ovoid, dorsoventrally compressed, and curves slightly posteroventrally from its base, while the medial condyle is broad, triangular, and protrudes slightly posterodorsally (*Figure 12A*). In ventral view, the condyles are separated by a broad, deep, and 'U'-shaped intercondylar groove (*Figure 12B*), that is confluent dorsally with a deep narrow sulcus that extends for a short distance on the posterior surface (*Figure 12A*). In ventral view, the lateral and medial condyles extend for approximately the same distance anteriorly and enclose a shallow anterior trough (*Figure 12B*). The lateral condyle has a mediolaterally narrow, subrectangular outline in distal view, and is inset from the lateral margin so that it is separated from it by a distinct notch (*Figure 12B and*

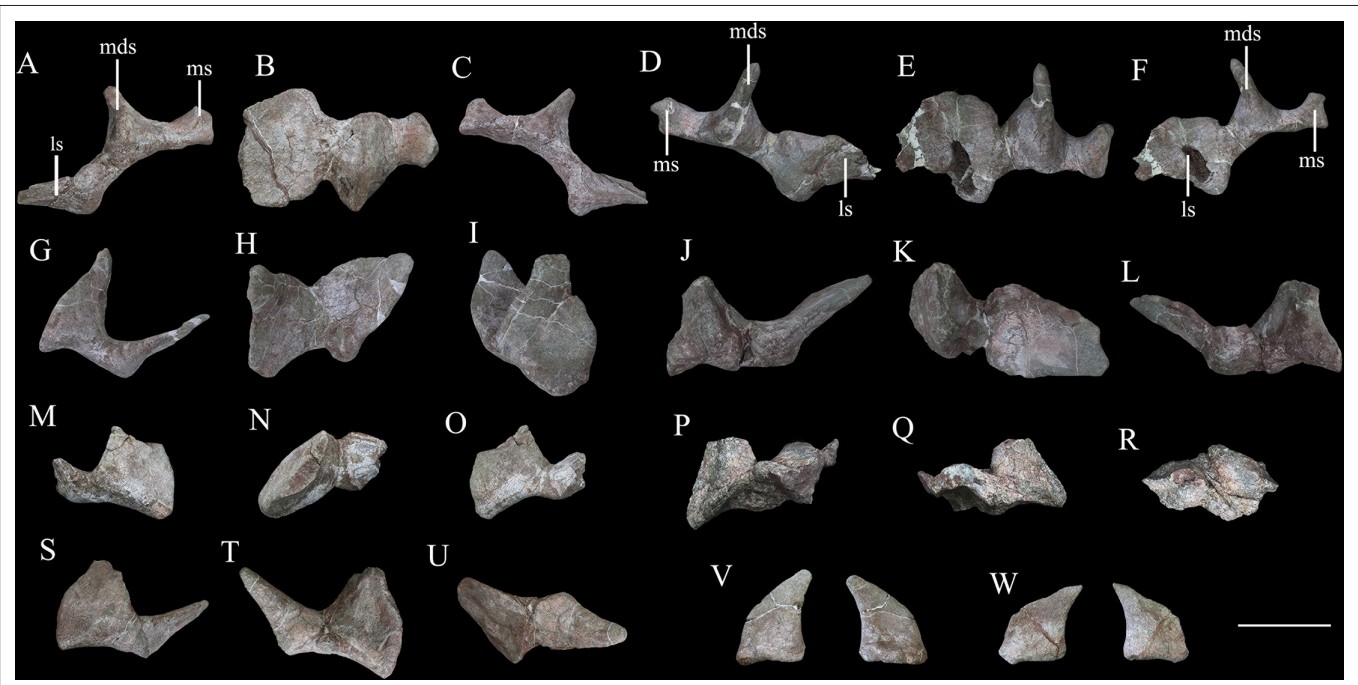

**Figure 13.** Cervical and pectoral osteoderms of *Yuxisaurus kopchicki*. Tripartite compound osteoderm (TPO) 1 in (**A**) anterior, (**B**) dorsal, and (**C**) posterior views. TPO 2 in (**D**) anterior, (**E**) dorsal, and (**F**) posterior views. Bipartite osteoderm (BPO) 1 in (**G**) anterior, (**H**) dorsal, and (**I**) medial views; BPO 2 in (**J**) anterior, (**K**) dorsal, and (**L**) posterior views; BPO 3 in (**M**) anterior, (**N**) dorsal, and (**O**) posterior views; BPO 4 in (**P**) anterior, (**Q**) posterior, and (**R**) dorsal views; and BPO 5 in (**S**) anterior, (**T**) posterior, and (**U**) dorsal views. Blade-like cervical spines in anterior and posterior views (**V, W**). Abbreviations: ls, lateral spine; mds, middle scute; ms, medial scute. Scale bar equals 10 cm.

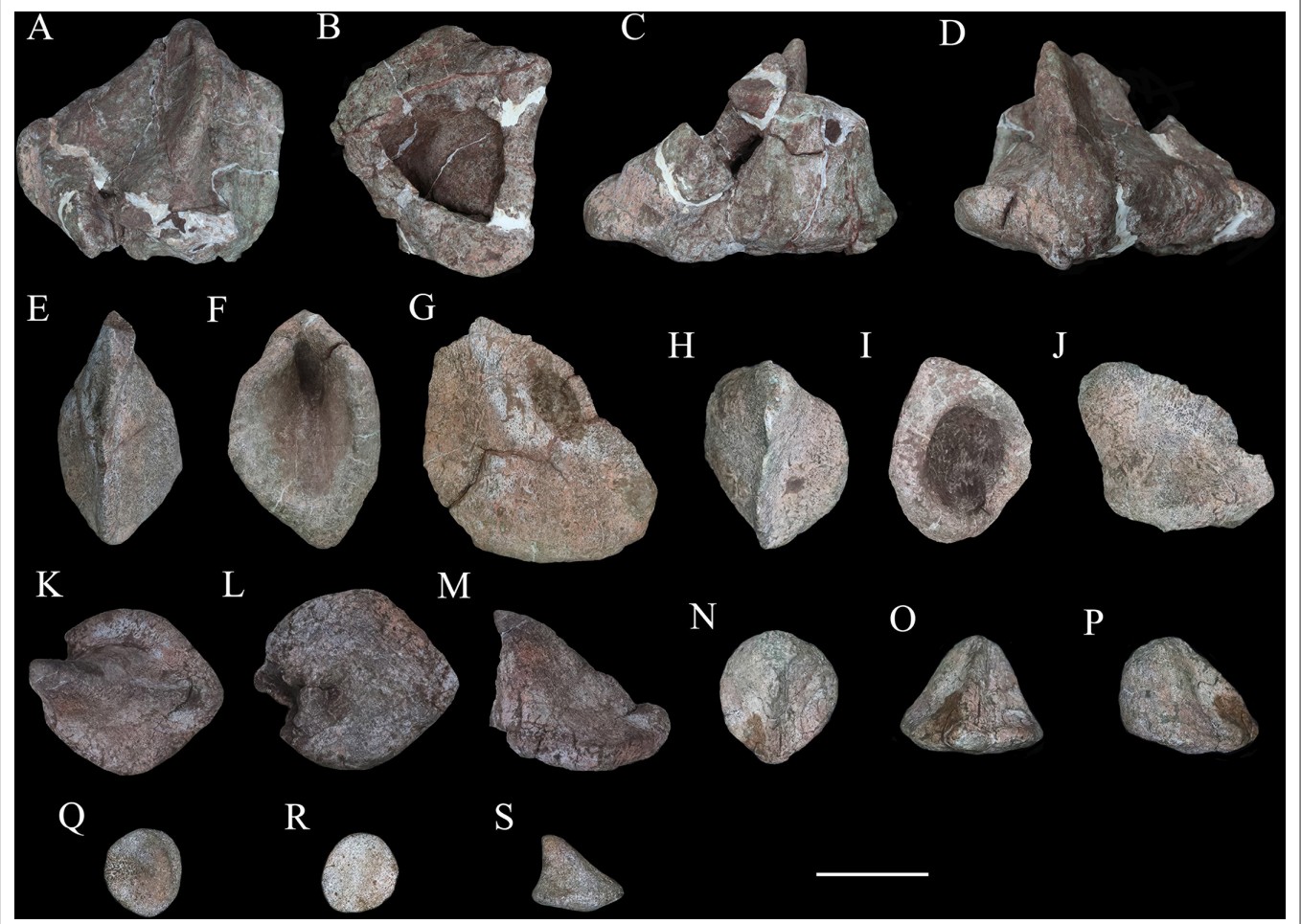

**Figure 14.** Six selected individual osteoderms of *Yuxisaurus kopchicki*. Osteoderm 1 in (**A**) dorsal, (**B**) ventral, (**C**) posterior, and (**D**) anterior views; osteoderm 2 in (**E**) dorsal, (**F**) ventral, and (**G**) lateral views; osteoderm 3 in (**H**) dorsal, (**I**) ventral, and (**J**) lateral views; osteoderm 4 in (**K**) dorsal, (**L**) ventral, and (**N**) lateral views; osteoderm 5 in (**N**) dorsal, (**O**) anterior, and (**P**) lateral views; osteoderm 6 in (**Q**) dorsal, (**R**) ventral, and (**S**) lateral views. Scale bar equals 5 cm.

*C*). The lateral condyle also projects slightly further posteriorly than the mediolaterally wider, rounded medial condyle. The border of the medial condyle is invaginated to form a broad, 'U'-shaped trough (*Figure 12B*), that is confluent with a shallow depression on the medial surface of the distal femur (*Figure 12D*). This trough/depression is absent in *Scelidosaurus* (*Norman, 2020a*), *Scutellosaurus* (*Colbert, 1981*; *Breeden et al., 2021*), ankylosaurs (e.g., *Kilbourne and Carpenter, 2005*; *Kirkland et al., 2013*; *Kinneer et al., 2016*), and stegosaurs (*Gilmore, 1914*), and is considered a potential autapomorphy of *Yuxisaurus*. A roughened swelling on the lateral surface just dorsal to the notch bounding the lateral condyle might represent the attachment of the M. gastrocnemius.

## Osteoderms

More than 120 osteoderms of *Y. kopchicki* were recovered (*Figure 13*, *Figure 14*, *Figure 15*). However, all of these were found disassociated, without direct evidence of their original life positions. Nevertheless, co-ossified osteoderms are usually present in the cervical or pectoral regions of thyreophorans whereas single osteoderms are distributed on other body parts (e.g., *Blows, 2001*; *Vickaryous and Russell, 2003*), allowing some tentative conclusions on their positions to be made.

### Cervical and pectoral osteoderms

Seven compound osteoderms are preserved. Two of these consist of three elements (tripartite osteoderms) and the remaining five consist of two elements (bipartite osteoderms) (*Figure 13*). In all of these

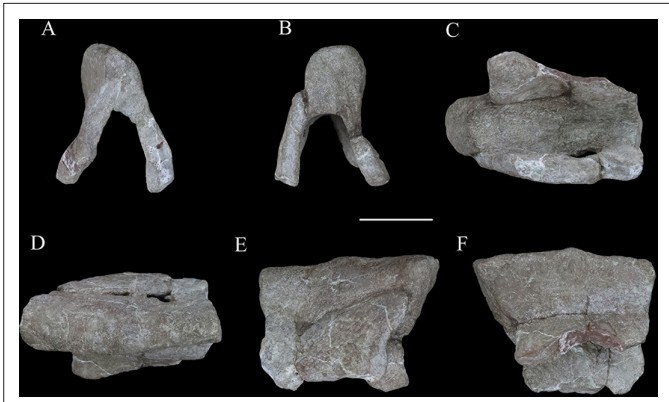

**Figure 15.** 'Pup tent'-shaped osteoderm of *Yuxisaurus kopchicki* in (**A**) posterior, (**B**) anterior, (**C**) ventral, (**D**) dorsal, and (**E, F**) side views. Scale bar equals 10 cm.

compound structures, the individual osteoderms are fused indistinguishably and it is likely that other co-ossified osteoderms were originally attached to some of these structures but were not preserved.

The two tripartite osteoderms (referred to hereafter as TPO 1 and 2) are similar in size and shape and mirror each other anatomically (*Figure 13A–F*). Each is composed of a blade-like lateral spine, a stouter, intermediate spine, and a conical medial osteoderm. In TPO 1 (*Figure 13A–C*), which is interpreted as from the right-hand side of the body, the base of the lateral spine is anteroposteriorly wide (128 mm) but thins dorsoventrally (45 mm). This spine extends laterally and its anterior and posterior margins are sharp. The straight anterior margin is inclined posteriorly while its posterior margin is slightly convex. Although the apex is missing, it seems reasonable to assume that the two edges converged to a point apically. Its lateral surface is swollen laterally in its central part. Four parallel ridges and the grooves between them extend on to the lateral surface from the base: however, these features are absent in TPO 2, which suggests that they might be due to accidental over-preparation. The intermediate spine of TPO 1 is directed dorsoventrally. It has a suboval base, which is anteroposteriorly elongated (108 mm) but transversely narrow (78 mm), and that is tall dorsoventrally (107 mm). In TPO 1, the anterior margin of the intermediate spine is long and convex, but in TPO 2 (inferred to be from the right-hand side; *Figure 13D–F*), this margin is divided into two straight edges. In both specimens, the posterior margins of these spines are deflected, and are consistently shorter than the anterior margins: as a result, the dorsal apex is posteriorly displaced relative to the base. Their lateral surfaces are concave and smooth, lacking foramina or grooves, and bear a central swelling, which is vertically directed, on either side. In both TPO 1 and 2, the medial-most osteoderm is the smallest of the three (*Figure 13A, C, D and F*). It is similar to the others, and in TPO 1 has an anteroposteriorly elongated (62 mm) but transversely narrow (40 mm) base. Nevertheless, the spine is more conical in shape with a smooth rounded lateral surface. It has a posteriorly displaced dorsal apex, which is almost flush with the posterior margin of the base. Its dorsal end bears a small protrusion.

In anterior or posterior views, the conjoined ventral surface of each tripartite structure is arched (*Figure 13A, C, D and F*), presumably corresponding to the neck shape of *Yuxisaurus*. Co-ossified cervical half-rings are present only in *Scelidosaurus* and ankylosaurians among Thyreophora (e.g., *Carpenter, 2001*; *Norman, 2021*), and vary in terms of the number of osteoderms included and their individual morphology (e.g., *Blows, 2001*). The partial cervical half-rings of *Yuxisaurus* are not fused to any other half-rings and closely resemble the third cervical half-ring of *Scelidosaurus*, as well as those of *Gargoyleosaurus*, *Sauropelta*, *Silvisaurus*, *Gastonia*, *Stegopelta*, and *Ankylosaurus* (*Carpenter, 1984*; *Ford, 2000*; *Kinneer et al., 2016*; *Norman, 2020b*). The external surfaces of the cervical osteoderms are generally smooth, similar to those of the ankylosaurians *Gastonia* and *Silvisaurus* (*Eaton, 1960*; *Kinneer et al., 2016*), whereas they are pitted or vascularized in the early branching thyreophorans *Scutellosaurus* and *Scelidosaurus*, and in most ankylosaurians, such as *Mymoorapelta*, *Edmontonia*, *Europelta*, *Saichania*, and *Pinacosaurus* (*Maryanska, 1977*; *Kirkland and Carpenter, 1994*; *Kirkland et al., 2013*; *Burns and Currie, 2014*; *Breeden and Rowe, 2020*; *Norman, 2020b*; *Breeden et al., 2021*).

The first bipartite osteoderm (termed 'BPO 1' hereafter) is composed of two spines (*Figure 13G–I*). These are similar in morphology, with an elongated oval base, a blade-like body and a convex dorsal end (*Figure 13G*). BPO 1 is inferred to be from the left side of the body when the spine apices are posteriorly placed relative to their base. The lateral spine curves dorsolaterally with a concave medial surface and convex lateral surface; the medial spine is straight with the lateral and medial surfaces nearly symmetrical to each other. The spine margins are somewhat convex, except that the posterior edge of the medial spine is straight. In posterior view, the lateral spine diverges from the medial spine at an angle of 45°. In dorsal view (*Figure 13H*), the lateral spine is more posteriorly placed than the medial spine, and the junction slightly narrows anteroposteriorly, leaving a broad concavity anteriorly and a narrow one posteriorly. Consequently, the lateral spine appears to contact the posterolateral portion of the medial spine. The ventral surface of the medial spine bears a curved ventromedial expansion, rendering its ventral surface concave. This expansion also extends anteriorly for a short distance.

The second bipartite osteoderm, BPO 2, has a similar configuration but is from the right-hand side of the body (*Figure 13J–L*). Compared to BPO 1, the medial spine is more robust (*Figure 13J*). This spine has a wide base and its distal end curves somewhat medially. The lateral spine diverges from the medial spine at an angle of 70° when the medial spine is vertically positioned. The surfaces of BPO 2 are not as smooth as those of BPO 1: grooves are present on the lateral surface of the lateral spine; the medial spine is medially pitted at the base; and the conjoined ventral surface is ornamented with striations. The junction between the individual osteoderms has suffered severe damage, leaving a large fissure.

A pair of symmetrical bipartite osteoderms, BPO 3 (*Figure 13M–O*) and BPO 4 (*Figure 13P–R*), are similar in size and appearance. Each is composed of two spines of distinct sizes. In both specimens, the larger spine is oval-based, has a nearly flat medial surface and a dorsoventrally concave but anteroposteriorly convex lateral surface. Both its anterior and posterior edges are curved and converge dorsally into a pointed apex. By contrast, the smaller spine has straight anterior and concave posterior margins that are both sharp, which terminate dorsally in a rounded apex. Its lateral surface is flat and the medial surface is convex. In dorsal view, the junction between the two osteoderms has a broad anterior concavity but an obtuse angle posteriorly, and unlike the condition in BPO 1, the lateral spine contacts the anterolateral corner of the medial spine. The co-ossified ventral surface is smooth and concave.

The fifth bipartite osteoderm (BPO 5) consists of a spine and a plate (*Figure 13S-U*). The spine has a long sharp edge, which extends obliquely and dorsally from the base, opposite to which is a short, blunt, vertical margin. Its lateral surfaces are strongly convex. The plate is generally flat dorsally, but half of it curves ventrally to meet, and project slightly beyond, the ventral margin of the spine. A gradual widening trough, which parallels part of the lateral surface, crosses the plate's dorsal surface and extends ventrally next to the spine along the curved half surface. The plate contacts the spine at the front of the short edge. The conjoined ventral surface is severely damaged.

To our knowledge, asymmetrical co-ossified bipartite osteoderms are present in the cervical armor of *Scelidosaurus*, *Hungarosaurus*, and *Struthiosaurus* (*Bunzel, 1871*; *Seeley, 1881*; *Pereda-Suberbiola and Galton, 2001*; *Ősi, 2005*; *Ősi et al., 2019*; *Norman, 2020b*), the lateral pectoral armor of *Edmontonia* and in a possible Early Jurassic ankylosaur from India (*Ford, 2000*; *Galton, 2019*). Therefore, these bipartite osteoderms were most likely from the cervical or pectoral region.

It seems likely that two isolated blade-like spines are also from the cervical region. These spines have an elongated oval base, so that the body and base are both narrow (*Figure 13V and W*). They both have a long convex edge and a short concave edge, so that the dorsal apex projects beyond the level of the base. The ventral half of the convex edge is nearly straight, then curves posterodorsally and continues dorsally with a mild curvature. The dorsal end is sharp on one spine but rounded on the other slightly larger one. Each spine has a depressed medial surface and a slightly convex lateral surface, although the larger spine (*Figure 13V*) bears a vertical depression on the convex surface close to its longest margin. The ventral surface is depressed but also bears an anteroventral expansion as in BPO 1 and BPO 2. These two spines are similar in appearance to the cervical spines of *Polacanthus* and the caudal plates of *Mymoorapelta* (*Kirkland and Carpenter, 1994*; *Blows and Honeysett, 2014*). However, the caudal plates are hollowed ventrally in *Mymoorapelta*; consequently, these spines are most likely from the cervical region of *Yuxisaurus*.

## Other osteoderms

Most other individual osteoderms are similar (*Figure 14*). They are oval-based, with a convex or slightly concave longest margin and a vertical or slightly concave short margin. These margins converge dorsally into an apex. Therefore, the body appears to be curved in osteoderms with a concave short margin, but straight in those with a vertical short margin. The longest margin is generally sharp whereas the shorter margin is rounded in some cases, although occasionally both margins are rounded. Ventrally they are generally flat but sometimes convex, with the ventral margins somewhat everted. The lateral surfaces are depressed, but generally bear a vertical swelling in their centers. It is noteworthy that 15 of the 120 osteoderms have a foramen or are excavated ventrally. Where present the foramina have rounded outlines and are usually small relative to the ventral surface area, but they appear to open out and expand into cavities within the osteoderm. By contrast, the ventral excavations are fully open, creating an osteoderm inner surface. Generally, the osteoderms with a solid ventral surface are smaller in size than those with a hollow base. The largest hollow-based osteoderm is damaged but was at least 160-mm long, 150-mm wide, and 110-mm tall (*Figure 14A–D*). With reference to *Scelidosaurus*, the relatively large hollow-based osteoderms probably formed the primary rows across the dorsolateral body surface or the caudal region, while other smaller osteoderms would have been interspersed among them (*Norman, 2020b*).

A unique 'pup tent'-shaped osteoderm is approximately 126-mm long and 94-mm tall but lacks anterior and posterior walls (*Figure 15*). It is triangular in cross-section and strongly excavated ventrally with a dorsal acute angle on one surface but rounded on the other side. Although weathered, the two buttresses are generally straight and divergent at an angle of ~48°. The outline between the buttresses resembles the overline outline of the osteoderm in both anterior and posterior views, and the external and inner surfaces are smooth and slightly depressed (*Figure 15*). In dorsal view, the roof is somewhat curved and the rounded end transversely expands more than the acute side. Viewed laterally, the dorsal roof is nearly straight, overhanging the ventral end on both sides. This osteoderm appears to be similar to an anterior median caudal scute referred to *Scelidosaurus* from Arizona, USA (*Padian, 1989*), and we propose that, in life, it was probably situated on the midline of the posterior part of the body of *Yuxisaurus*. Alternatively, this unusual morphology might represent a pathology.

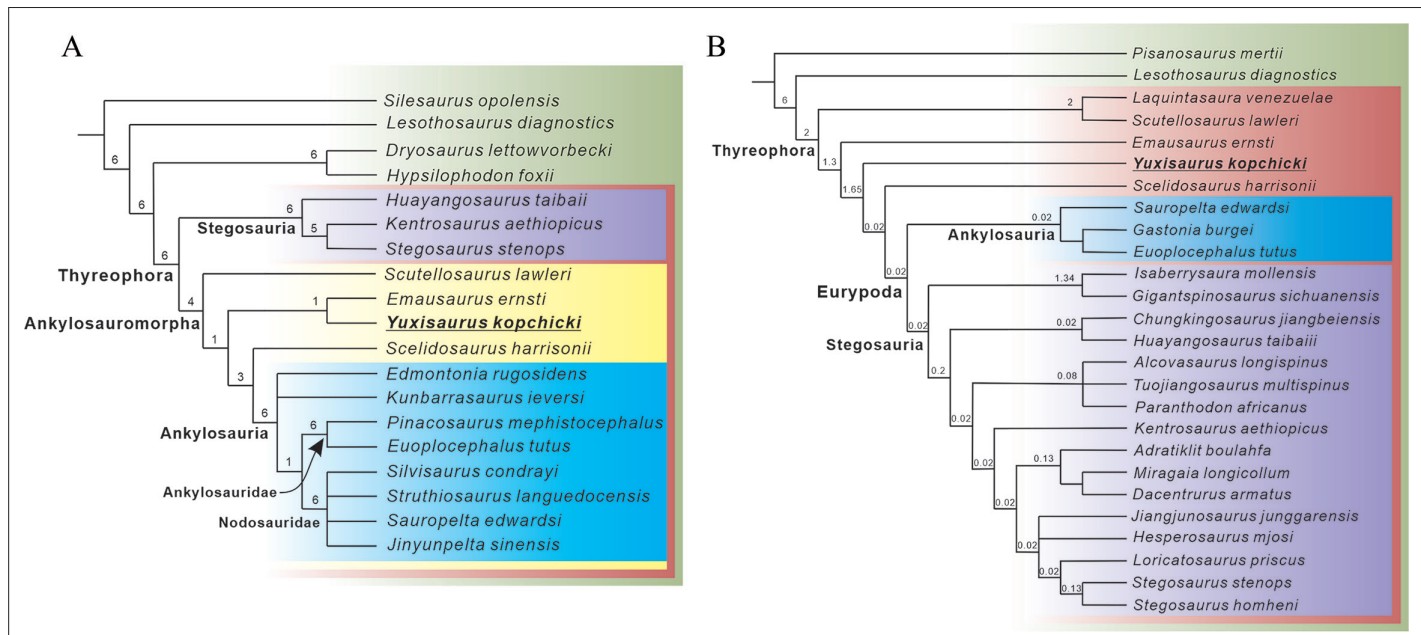

**Figure 16.** Phylogenetic relationships of *Yuxisaurus* within Thyreophora. (**A**) Strict consensus of the six most parsimonious trees (MPTs) recovered from analysis of the modified *Norman, 2021* data set. (**B**) Strict consensus of the two MPTs recovered from analysis of the modified *Maidment et al., 2020* data set. Bremer support values are shown adjacent to the nodes.

## Discussion

### Phylogenetic results

Analysis of the *Norman, 2021* data set resulted in the recovery of the six most parsimonious trees (MPTs) with tree lengths of 238 steps, a Consistency Index of 0.710 and a Retention Index of 0.858. A strict consensus of these trees is shown in *Figure 16A*. This analysis recovered *Scutellosaurus* as the earliest diverging member of a clade that also includes *Emausaurus*, *Yuxisaurus*, *Scelidosaurus*, and ankylosaurs; and this clade is the sister group of stegosaurs. *Y. kopchicki* is found in a clade with *Emausaurus* from the Toarcian of Germany, but support for this clade is weak (Bremer value of 1). This unnamed clade is in turn the sister-group of *Scelidosaurus* + Ankylosauria.

Inclusion of *Yuxisaurus* within the (*Scutellosaurus* (*Emausaurus*+ *Yuxisaurus*) (*Scelidosaurus* + Ankylosauria)) clade is supported by the possession of the following unambiguous synapomorphies: 13 (1), cranial exostoses (cortical bone ornamentation) present; 17 (1), remodeling of the external surface of skull bones partial; 105 (1), osteoderms form parasagittal rows either side of dorsal midline; 108 (1), lateral flank osteoderms ovoid and keeled; and 117 (1), cervical osteoderms present. The unnamed clade including *Y. kopchicki* and *E. ernsti* is supported by a single synapomorphy: 36 (1), basipterygoid process posteroventrolaterally oriented. This clade lacks the three synapomorphies uniting *Scelidosaurus* + Ankylosauria, namely: 14 (1), skull (non-supraorbital) osteoderms present; 17 (2 and 3), remodeling of the external surface of skull bones partial with few osteoderms or extensively osteoderm covered; and 18 (1), postorbital(non-supraorbital) osteoderms present.

The analysis based on the *Maidment et al., 2020* data set produced two MPTs with tree lengths of 269 steps, a Consistency Index of 0.605 and a Retention Index of 0.663. A strict consensus of the trees is shown in *Figure 16B*. This analysis recovered *Yuxisaurus* within Thyreophora, as an early diverging branch between *Emausaurus* and *Scelidosaurus*. Ankylosauromorpha (neither sensu *Carpenter, 2001* nor sensu *Norman, 2021*; see below) was not recovered, *Scutellosaurus* and *Emausaurus* were found outside of Eurypoda, and the *Emausaurus* + *Yuxisaurus* clade was not identified. *Yuxisaurus* has a single unambiguous synapomorphy of Thyreophora: 29 (1) maxillary tooth row inset medially from the lateral surface. It is grouped with *Scelidosaurus* and Eurypoda to the exclusion of *Emausaurus* in having the following synapomorphies: 32 (1) supraorbital elements form the dorsal rim of the orbit; and 110 (1) 'U'-shaped cervical collars composed of keeled scutes present. *Yuxisaurus* is excluded from the *Scelidosaurus* + Eurypoda clade as it lacks the unambiguous synapomorphy of the latter group: 57 (0) cervical vertebrae longer anteroposteriorly than wide transversely.

### Comments on Ankylosauromorpha

*Carpenter, 2001* conducted a phylogenetic analysis that recovered a monophyletic Eurypoda split into two sister lineages, Stegosauria and *Scelidosaurus* + Ankylosauria, with *Emausaurus* and *Scutellosaurus* as successive outgroups to Eurypoda. This result contrasted with previous results where *Scelidosaurus* was excluded from Eurypoda (e.g., *Sereno, 1986*; *Sereno, 1999*). To recognize the new *Scelidosaurus* + Ankylosauria clade *Carpenter, 2001* (p. 471) proposed the name Ankylosauromorpha, which he defined thus: 'Ankylosauromorpha are thyreophorans that are closer to *Scelidosaurus*, *Minmi*, Polacanthidae, Nodosauridae, and Ankylosauridae, than to *Stegosaurus*'. However, the 'ankylosauromorph hypothesis,' was not supported by later analyses, which failed to reproduce this result and consistently placed *Scelidosaurus* outside Eurypoda (e.g., *Norman et al., 2004*; *Butler et al., 2008*; *Boyd, 2015*; *Dieudonné et al., 2021*).

Subsequently, a new phylogenetic analysis by *Norman, 2021* provided additional support for a sister-group relationship between *Scelidosaurus* and Ankylosauria. However, this analysis also recovered *Emausaurus* and *Scutellosaurus* as outgroups to this clade, with all of these taxa more closely related to ankylosaurs than stegosaurs. This prompted *Norman, 2021* to expand the *Carpenter, 2001* ankylosauromorph concept to encompass these additional taxa, even though the latter author did not include them within his original definition. The (*Norman, 2021*) new definition for Ankylosauromorpha was: 'All taxa more closely related to *Euoplocephalus* and *Edmontonia* than to *Stegosaurus*.' However, this definition is functionally identical to the existing stem-based definitions of Ankylosauria provided by *Carpenter, 1997*: 'All thyreophoran ornithischians closer to *Ankylosaurus* than to *Stegosaurus*' and *Sereno, 1998*: 'All eurypods closer to *Ankylosaurus* than *Stegosaurus*'. Hence, the tree topology provided by *Norman, 2021* implies that *Scutellosaurus*, *Emausaurus*, *Yuxisaurus*, and *Scelidosaurus* are ankylosaurs under these previous and broadly applied phylogenetic

definitions; consequently, the *Norman, 2021* stem-based use of 'Ankylosauromorpha' is in error and his redefinition of the clade redundant. However, if the *Norman, 2021* topology were to receive further support in future, a case could be made for a node-based definition of Ankylosauromorpha (e.g., a clade consisting of *Scelidosaurus*, *Ankylosaurus*, their common ancestor and all of its descendants) or some other variation.

## Thyreophoran biogeography

The discovery of *Yuxisaurus* cements the presence of armored dinosaurs in the Early Jurassic of Eastern Asia, an observation previously supported by the fragmentary material assigned to '*Tatisaurus*' and '*Bienosaurus*' (*Simmons, 1965*; *Dong, 2001*). The inadequate holotype specimens of the latter taxa do not allow them to be incorporated into formal phylogenetic or macroevolutionary analyses (*Norman et al., 2007*; *Raven et al., 2019*), and the only other Early Jurassic thyreophoran material reported from Asia—from the Kota Formation of India (*Nath et al., 2002*; *Galton, 2019*)—is also frustratingly incomplete (and might be of Middle Jurassic age: *Prasad and Parmar, 2020*). Moreover, it is worth noting that India was part of Gondwana during the Jurassic and so its fauna would likely have been biogeographically distinct from that occurring in China, which formed part of Laurasia. Hence, it has been impossible to include any early diverging Asian taxa in broad-scale tree-based analyses of early thyreophoran evolutionary history thus far. However, the more complete, and highly distinctive, material of *Yuxisaurus* enables some more substantive discussion of these issues.

For example, until relatively recently all of the valid Early Jurassic thyreophoran taxa included in such analyses were from North America (*Scutellosaurus*) or Europe (*Emausaurus* and *Scelidosaurus*) limiting our ability to determine their biogeographic history beyond suggesting a Laurasian distribution (e.g., *Sereno, 1999*; *Norman et al., 2004*). However, new phylogenetic analyses have proposed that two other taxa, *Lesothosaurus* and *Laquintasaura*, might be early members of Thyreophora (e.g., *Butler et al., 2008*; *Boyd, 2015*; *Baron et al., 2017a*; *Raven and Maidment, 2017*; *Maidment et al., 2020*), although these views are contentious and alternative relationships for these taxa have been posited (e.g., *Dieudonné et al., 2021*; *Barta and Norell, 2021*; *Norman, 2021*). If *Lesothosaurus* and *Laquintasaura* are thyreophorans, however, this broadens the palaeogeographic distribution of the clade to Gondwana in the earliest Jurassic, with *Laquintasaura* from the Hettangian of Venezuela (*Barrett et al., 2014*) and *Lesothosaurus* from the Sinemurian of southern Africa (*Viglietti et al., 2020*), implying that the group might have originated in Gondwana and dispersed to Laurasia (*Boyd, 2015*; *Raven et al., 2019*; *Maidment et al., 2020*). If it is of the Early Jurassic age, the material from the Kota Formation would support an early Gondwanan distribution also (*Nath et al., 2002*; *Galton, 2019*).

The two phylogenetic analyses, we selected to assess the relationships of *Yuxisaurus* reflect differing opinions on the relationships of *Lesothosaurus* and underscore current uncertainties in early ornithischian biogeography. In our iteration of the *Norman, 2021* analysis (see Results, above), *Lesothosaurus* is recovered as a non-thyreophoran ornithischian and, as a result, *Yuxisaurus* belongs to a grade of early diverging thyreophoran taxa whose entire early evolutionary history is confined to Laurasia. This scenario implies that all later-occurring Gondwanan taxa were dispersals from Eurasia. The sister-group relationship of *Yuxisaurus* and *Emausaurus* implies a pan-Eurasian distribution for this small clade, but taken with the North American distribution of the earlier-diverging *Scutellosaurus* and the European occurrence of the later-diverging *Scelidosaurus*, there is no clear biogeographic signal within the broader Laurasian region. By contrast, the tree topology gained from analysis of the *Maidment et al., 2020* data set (see Results, above) recovers *Laquintasaura* as a thyreophoran, which might imply a greater role for Gondwana in the origin of the group although it still recovers *Yuxisaurus* as a member of a primarily Laurasian radiation. Unfortunately, the lack of consensus on early ornithischian phylogeny prevents us from choosing between these scenarios: specimens from currently unsampled areas, new anatomical data, and agreement on character coding and scoring decisions will be required to move this debate forwards.

Minimally, however, the recognition of *Yuxisaurus* further highlights that thyreophorans achieved a global (or at least pan-Laurasian) distribution rapidly during their early history, perhaps in the space of only 2–3 million years (up to a maximum of ~10 Ma) (see also *Raven et al., 2019*). This time scale is suggested by the current absence of Triassic ornithischians (unless silesaurids are considered members of this clade: *Müller and Garcia, 2020*) and the occurrences of the earliest diverging members of

Thyreophora, which all have potential first appearance dates ranging from Hettangian–Sinemurian (201.3–190.8 Ma: *Walker et al., 2018*).

Early thyreophorans have been recovered in a diverse range of palaeoenvironmental and taphonomic settings and as components of remarkably different ecosystems, suggesting that their early radiation might have been underpinned by greater ecological diversity among them than usually appreciated. For example, *Lesothosaurus*, *Laquintasaura,* and *Scutellosaurus* were likely obligate bipeds (*Thulborn, 1972*; *Colbert, 1981*; *Barrett et al., 2014*), whereas the more heavily built *Yuxisaurus* and *Scelidosaurus* were likely quadrupeds (*Maidment et al., 2014*; *Norman, 2021*). Moreover, there is some evidence of dietary variation with the possibility that *Lesothosaurus* was a facultative omnivore (*Barrett, 2000*), whereas *Scelidosaurus* is thought to have been an obligate herbivore (*Barrett, 2001*; *Norman, 2021*). Early members of the clade, like *Lesothosaurus* and *Laquintasaura*, were apparently unarmored (*Thulborn, 1972*; *Barrett et al., 2014*), but armor became a conspicuous feature of all later-diverging members of the group (*Norman et al., 2004*) and varied considerably even in the earliest appearing taxa (*Colbert, 1981*; *Norman, 2020b*; this paper). Several early experiments in sociality and group-living are inferred based on mass accumulations of several taxa (*Barrett et al., 2014*; *Barrett et al., 2016*). In terms of habitats, *Emausaurus* and *Scelidosaurus* are known from marine settings (*Haubold, 1990*; *Norman, 2020c*), suggesting that they lived in low-lying well-watered coastal areas, but other taxa, such as *Lesothosaurus* are known from settings that were far inland and at least seasonally arid (*Viglietti et al., 2020*). Finally, several thyreophorans represent the most abundant dinosaur taxa known from their respective formations (e.g., *Scelidosaurus*, *Scutellosaurus,* and *Laquintasaura* which are each represented by multiple specimens: *Colbert, 1981*; *Barrett et al., 2014*; *Norman, 2020c*; *Breeden et al., 2021*), but in other cases, they seem to be subordinate components of their ecosystems (e.g., *Lesothosaurus* is known from multiple specimens but is much less abundant than the sauropodomorph dinosaurs from the upper Elliot Formation: *Knoll, 2005*; *Viglietti et al., 2020*) or rather rare (e.g., *Yuxisaurus*, which also occurs in a sauropodomorph-dominated fauna: *Mao et al., 2019*).

## Conclusions

A partial skeleton collected from the Lower Jurassic Fengjiahe Formation of Yunnan Province, China, represents a new taxon of early diverging thyreophoran dinosaur, which we name *Y. kopchicki* (*Figure 17*). It can be distinguished from all other thyreophorans by a suite of autapomorphic cranial, axial and appendicular character states, as well as a unique combination of character states. *Yuxisaurus* represents the first unambiguous armored dinosaur to be recovered from the Lower Jurassic of Asia that is based on associated, diagnostic material and is the first that is complete enough to be incorporated into a phylogenetic analysis. Although its relationships are heavily dependent on the preferred data set, our analyses recover *Yuxisaurus* as an outgroup to either *Scelidosaurus* + Ankylosauria or *Scelidosaurus* + Eurypoda, with the former analysis also suggesting a sister-group relationship to the European taxon *Emausaurus*. *Yuxisaurus* helps to emphasize the pan-Laurasian (and

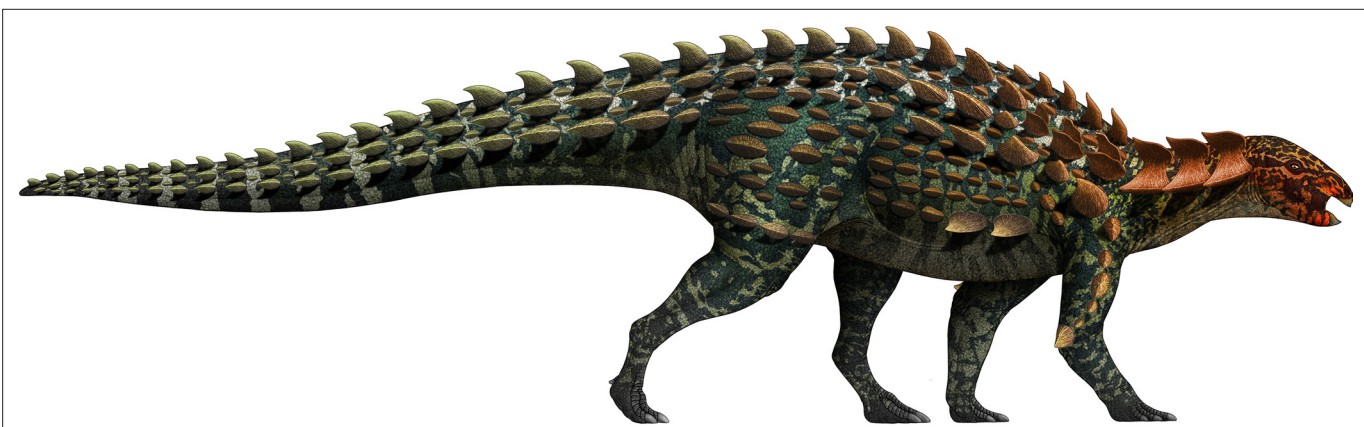

**Figure 17.** Life restoration of *Yuxisaurus kopchicki*. The osteoderm arrangement is hypothetical but that includes many of the types of armor found with the skeleton.

possibly global) distribution of early thyreophorans, their diverse morphology and ecology, and the rapidity of their initial radiation.

# Materials and methods
## Phylogenetic analysis

In order to investigate the phylogenetic position of *Y. kopchicki,* it was scored into two recently published data matrices incorporating other early diverging thyreophorans (*Maidment et al., 2020*; *Norman, 2021*) that differ in their taxonomic coverage and in the relationships recovered among these taxa.

Norman's (2021) original analysis included 18 taxa scored for 115 characters. We added three new characters: 116, lacrimal ramus of jugal directed horizontally (0) or posteroventrally (1); 117, cervical osteoderms, absent (0), present (1); and 118, surface texture of osteoderms, pitted (0) or smooth (1). With the addition of *Yuxisaurus*, this resulted in a data set composed of 19 taxa and 118 characters. The data matrix was compiled in Mesquite v. 2.72 (*Maddison and Maddison, 2007*) and was analyzed using TNT v. 1.1 (*Goloboff et al., 2008*). Following the protocols in *Norman, 2021*, *Silesaurus opolensis* was designated as the outgroup and all characters were of equal weight and unordered. *Norman, 2021* analyzed his data using both Branch and Bound and heuristic searches with Tree Bisection-Reconnection in PAUP, whereas our analysis used a 'traditional' heuristic search with one random seed and 1000 replicates of Wagner trees.

When scores for *Y. kopchicki* were added to the *Maidment et al., 2020* matrix this resulted in a data set composed of 26 taxa and 115 morphological characters. The matrix was analyzed in TNT v. 1.1 using 'traditional' heuristic search with one random seed and 1000 replicates of Wagner trees. Following the original settings used in *Maidment et al., 2020*, *Pisanosaurus* was assigned as the outgroup, and all characters were equally weighted, and characters 105 and 106 were ordered, as were the continuous characters (characters 1–24).

Bremer supports were calculated for both analyses by sequentially increasing the search depth to a maximum hold of 11,000 optimal trees and 6 suboptimal trees in memory to test the robustness of each node.

# Acknowledgements

The authors would like to thank the field crew of Yimen Administration of Cultural Heritage for their efforts in the discovery, excavation, and preliminary preparation of this specimen. The authors thank X Hou and Y He for assistance; Z Yang for preparation; Y Chen and R Jiang for help with illustration; and H Xing and S Maidment for suggestions and discussion. The authors are grateful to the reviewers Attila Ősi and Benjamin Breeden III for their constructive critiques that greatly improved the manuscript. The phylogenetic analysis software TNT version 1.1 was provided by generosity of the Willi Hennig Society. Support for this research is from the Double First-Class joint program of Science & Technology Department of Yunnan and Yunnan University (2018FY001-005).

# Additional information

### Funding

| Funder | Grant reference number | Author |
| --- | --- | --- |
| Double First-Class joint program of Science & Technology Department of Yunnan and Yunnan University | 2018FY001-005 | Shundong Bi |

The funders had no role in study design, data collection and interpretation, or the decision to submit the work for publication.

### Author contributions
Xi Yao, Formal analysis, Investigation, Methodology, Software, Validation, Visualization, Writing – original draft; Paul M Barrett, Formal analysis, Investigation, Methodology, Validation, Writing – original draft; Lei Yang, Resources; Xing Xu, Conceptualization, Investigation, Writing - review and editing; Shundong Bi, Conceptualization, Data curation, Formal analysis, Funding acquisition, Methodology, Project administration, Resources, Supervision, Validation, Visualization, Writing - review and editing

### Author ORCIDs
Paul M Barrett (ID) http://orcid.org/0000-0003-0412-3000
Xing Xu (ID) http://orcid.org/0000-0002-4786-9948
Shundong Bi (ID) http://orcid.org/0000-0002-0620-187X

### Decision letter and Author response
Decision letter https://doi.org/10.7554/eLife.75248.sa1
Author response https://doi.org/10.7554/eLife.75248.sa2

---

# Additional files

### Supplementary files
• Supplementary file 1. Data matrix modified from *Norman, 2021* used in the phylogenetic analysis (in txt format).

• Supplementary file 2. Data matrix modified from *Maidment et al., 2020* used in the phylogenetic analysis (in txt format).

• Transparent reporting form

### Data availability
All data generated or analysed during this study are included in the manuscript and Supplementary Information.

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
