## [Editor Report]

This paper reports a new species of armored dinosaur from rocks in southwestern China dated to the beginning of the Jurassic Period. This represents the first valid species of armored dinosaur from the Early Jurassic in Asia, as although the presence of armored dinosaurs in Asia has been documented for decades based on isolated jaw bones referred to Thyreophora-the group of armored dinosaurs-none that material was complete enough for diagnosis to a known or new species. This new specimen demonstrates the rapid diversification and distribution of armored dinosaurs across the northern hemisphere early in their evolutionary history.

---

## [Decision Letter]

**Decision letter after peer review:**

Thank you for submitting your article "A new early-branching armoured dinosaur from the Lower Jurassic of southwestern China" for consideration by *eLife*. Your article has been reviewed by 2 peer reviewers, and the evaluation has been overseen by a Reviewing Editor and George Perry as the Senior Editor. The following individuals involved in review of your submission have agreed to reveal their identity: Attila Ősi (Reviewer #1); Benjamin Breeden (Reviewer #2).

The two reviewers were strongly aligned in their positive views about your study and manuscript and their suggestions for your revision are both good and manageable, so I include them below in their entirety to address in your revision. Well done!

*Reviewer #1 (Recommendations for the authors):*

1) Comments to the text:

– The osteological description is thorough, the text clear and easy to follow. I found hardly any typos or spelling mistakes.

– l. 194: Subtitle „ Skull" should be „ Skull and mandible" since the rest of the two hemi-mandibles are also described.

– l. 202: „An anteroposteriorly elongated antorbital fossa excavates its lateral surface deeply. The antorbital fossa is rounded and subtriangular in outline". So how is this antorbital fossa?

– l. 213: „Yuxisaurus also appears to lack the anterior antorbital fenestra present in Scelidosaurus (Norman, 2020a), but this area is still encased in matrix." and then in l. 235: „The antorbital fenestra is a rounded opening in medial view." – I guess you mean here antorbital foramen as indicated on Figure 1. Also check l. 271.

– l. 290. The nomenclature of the “Supraorbitals" is a bit confusing. Use either “palpebral" or “supraorbital" along the text. In Figure 3 is “mso, mesosupraorbital" corresponds to middle supraorbital in the text (l. 294, 296)?

– l. 521. In the subtitle, I suggest to write Mandible or Hemi-mandibles instead of Mandibles.

– In the description of the mandible fragments it would be good to see some comparison with the post-dentary remains of Sarcolestes from the late Middle Jurassic of England (Galton 1983) as well. That is a well preserved specimen, already with some lateral ornamentation. I see that the authors write that the lateral surface is smooth (l. 533), but on Figure 7A it seems that this surface is somehow irregular. Is this only due to taphonomic processes or perhaps some mildly ornamentation can be detected?

– l. 897: change „((C))" to (C).

– l. 974: “the lateral spine is more posteriorly placed than the lateral spine". You mean: …than the medial spine?

– l. 1006-1008: „Asymmetrical co-ossified bipartite osteoderms are uncommon and present only among the cervical armour of Scelidosaurus, the lateral pectoral armour of Edmontonia and in a possible Early Jurassic ankylosaur from India". In addition it is also present in Struthiosaurus (Bunzel 1871, Seeley, 1881, Pereda-Suberbiola and Galton 2001) and Hungarosaurus (Ősi 2005, Ősi et al., 2019).

– A short summary of how many belts the neck armour might have consisted of, and how this is similar/different from the neck armour of Scelidosaurus and more advanced ankylosauromorphs (e.g. Gargoyleosaurus, Saurpelta, Hungarosaurus), would be really useful.

– l. 1242: suggest to add Botfalvai et al., (2021) dealing with the possible gregarious behaviour of ankylosaurs.

– l. 1261-1262 „Yuxisaurus represents the first armoured dinosaur to be recovered from Asia that is based on associated, diagnostic material”. I suggest to either change „the earliest armoured dinosaur…” or add „the first unambiguous armoured dinosaur to be recovered from the Lower Jurassic of Asia”

2) Comments to figures:

– In general, figures are detailed and of good resolution.

– In Figure 1 the stratigraphic column should be a bit larger using a larger font size.

– On Figure 8. ‘ard’ is not explained.

3) Citation – reference is complete.

– Change „Brandon and Carpenter 2005” to Kilbourne and Carpenter 2005

– l. 1167: Sereno 1998 is Sereno 1999?

*Reviewer #2 (Recommendations for the authors):*

I was delighted to be asked to review this manuscript after seeing the preprint circulating online a few months back. This is a very exciting specimen. Given how relatively abundant saurischian material is in the Lufeng and Fengjiahe formations, it was frustrating that the corresponding ornithischian record was so sparse! Finally China-and indeed Laurasian Asia!-has a good diagnostic thyreophoran from the Early Jurassic.

In my public review, I mentioned that adding Yuxisaurus to a dataset with a broader sampling of basal non-thyreophoran ornithischian taxa (e.g., Han et al., 2018 – J. Syst. Palaeontol. Or a derivative dataset) would strengthen the study; however, I realize this would add a lot of work, and I don’t think it’s entirely necessary as you have convincingly demonstrated that the specimen is a thyreophoran, and the aim of your study was not to clarify character polarization in early ornithischian taxa.

I was able to download the two supplemental data files containing the modified phylogenetic matrices from Norman (2021) and Maidment et al., (2020). Although I was able to replicate the authors' results using the modified Norman matrix, as detailed below, Supplementary file 2 (the modified Maidment et al., dataset) differs from the matrix described in text and needs to be replaced with the correct matrix prior to publication.

---

## [Author Response]

Reviewer #1 (Recommendations for the authors):1) Comments to the text:– The osteological description is thorough, the text clear and easy to follow. I found hardly any typos or spelling mistakes.

Thank you for the positive comments.

– l. 194: Subtitle „ Skull" should be „ Skull and mandible" since the rest of the two hemi-mandibles are also described.

Corrected.

– l. 202: „An anteroposteriorly elongated antorbital fossa excavates its lateral surface deeply. The antorbital fossa is rounded and subtriangular in outline". So how is this antorbital fossa?

Our descriptions of these features are accurate. The antorbital fossa is an excavation anterior to the orbit. This excavation has the morphology discussed in the text. By definition, a fossa is blind-ending pocket - not a perforation. The antorbital fossa houses (and is sometimes replaced by) a perforation that passes through the maxilla, which is termed the antorbital fenestra or foramen. The distinction we capture in our description is the difference between the depressed area (the fossa) and the actual hole (foramen/fenestra) that is housed in it. This distinction is important anatomically and captured both in phylogenetic data matrices and many detailed descriptions of dinosaur anatomy. In this case, the fossa has a broad, rounded subtriangular outline. It contains, in its posterior part, a small round antorbital fenestra. To remove potential confusion, we change ‘foramen’ to ‘fenestra’ where needed in the figure captions and text.

– l. 213: „Yuxisaurus also appears to lack the anterior antorbital fenestra present in Scelidosaurus (Norman, 2020a), but this area is still encased in matrix." and then in l. 235: „The antorbital fenestra is a rounded opening in medial view." – I guess you mean here antorbital foramen as indicated on Figure 1. Also check l. 271.

We agree with the reviewer that the nomenclature here is a bit confusing. Following Norman (2020), and many anatomical studies on other archosaurs, we use terms anterior antorbital foramen for an anteriorly positioned accessory foramen within the antorbital fossa and the term antorbital fenestra (which occurs infrequently within the clade) to refer to the posteriorly positioned major opening in the antorbital fossa (which is a diagnostic feature of the clade). Thus we changed the text of line 215 to “*Yuxisaurus* also appears to lack the anterior antorbital foramen”. In Fig. 3, we used ‘afo’ to refer to ‘antorbital fossa’ and ‘af’ to refer to ‘antorbital fenestra’.

– l. 290. The nomenclature of the “Supraorbitals" is a bit confusing. Use either “palpebral" or “supraorbital" along the text. In Figure 3 is “mso, mesosupraorbital" corresponds to middle supraorbital in the text (l. 294, 296)?

We agree and given that thyreophorans have more supraorbitals than other ornithischians we have changed the terminology accordingly. We now refer to the palpebral as the anterior supraorbital (noting that this homologous with the palpebral of other ornithischians) and the second supraorbital as the mesosupraorbital. Changes made to Figure 3 and throughout the text as necessary.

– l. 521. In the subtitle, I suggest to write Mandible or Hemi-mandibles instead of Mandibles.

Corrected, line 521 ‘Mandibles’ changed to ‘Mandible’.

– In the description of the mandible fragments it would be good to see some comparison with the post-dentary remains of Sarcolestes from the late Middle Jurassic of England (Galton 1983) as well. That is a well preserved specimen, already with some lateral ornamentation. I see that the authors write that the lateral surface is smooth (l. 533), but on Figure 7A it seems that this surface is somehow irregular. Is this only due to taphonomic processes or perhaps some mildly ornamentation can be detected?

Sarcolestes has a typical ankylosaur mandible in possessing a fused plate on the lateral surface, and it also differs from Yuxisaurus in having a high articular surface for the quadrate represented by a deep retroarticular process. Moreover, the lateral surface of the post-dentary hemimandible of Yuxisaurus is smooth with no ornamentation contrasting with sculpturing lateral surface of the Sarcolestes mandible (Galton, 1983, Armored dinosaurs (Ornithischia: Ankylosauria) from the Middle and Upper Jurassic of Europe). The irregular surface in the picture may be due to taphonomic processes. Yuxisaurus can be clearly distinguished from Sarcolestes by the above features. To clarify this feature we have added the following sentence: “Their lateral surfaces are smooth and bear no ornamentation or fused osteoderms.”

– l. 897: change „((C))" to (C).

Changed

– l. 974: “the lateral spine is more posteriorly placed than the lateral spine". You mean: …than the medial spine?

Changed to ‘… than the medial spine’

– l. 1006-1008: „Asymmetrical co-ossified bipartite osteoderms are uncommon and present only among the cervical armour of Scelidosaurus, the lateral pectoral armour of Edmontonia and in a possible Early Jurassic ankylosaur from India". In addition it is also present in Struthiosaurus (Bunzel 1871, Seeley, 1881, Pereda-Suberbiola and Galton 2001) and Hungarosaurus (Ősi 2005, Ősi et al., 2019).

*Hungarosaurus* and *Struthiosaurus* do indeed have these asymmetrical co-ossified bipartite osteoderms, thank you for pointing this out, and we have added this information to the text along with relevant references suggested.

– A short summary of how many belts the neck armour might have consisted of, and how this is similar/different from the neck armour of Scelidosaurus and more advanced ankylosauromorphs (e.g. Gargoyleosaurus, Saurpelta, Hungarosaurus), would be really useful.

We are uncertain of the number of cervical half-rings and would prefer not to speculate upon this in the paper. The two tripartite osteoderms could have comprised one complete cervical half-ring or represent different half-rings. The five bipartite osteoderms are more likely from the shoulder or pectoral region, but could also represent parts of cervical half-rings. So the number present could have been anywhere between 1 and 7, which covers the range of variation in cervical half-ring numbers present in other thyreophorans, such as *Scelidosaurus* (5), *Gargoyleosaurus* (2), *Sauropelta* (3), and *Hungarosaurus* (2). Given this ambiguity, we would prefer not to make any significant comparisons between *Yuxisaurus* and other thyreophorans in this respect.

– l. 1242: suggest to add Botfalvai et al., (2021) dealing with the possible gregarious behaviour of ankylosaurs.

Botfalvai et al. (2021) focuses on the gregarious behaviour of Cretaceous ankylosaurs, thus we consider it inappropriate in this context, which is focused on early thyreophorans.

– l. 1261-1262 „Yuxisaurus represents the first armoured dinosaur to be recovered from Asia that is based on associated, diagnostic material”. I suggest to either change „the earliest armoured dinosaur…” or add „the first unambiguous armoured dinosaur to be recovered from the Lower Jurassic of Asia”

line 1261-1262 changed to ‘*Yuxisaurus* represents the first unambiguous armoured dinosaur to be recovered from the Lower Jurassic of Asia…’.

2) Comments to figures:– In general, figures are detailed and of good resolution.

Thank you for the positive feedback.

– In Figure 1 the stratigraphic column should be a bit larger using a larger font size.

We adjusted the size of the stratigraphic column in Figure 1 and used a larger font size as suggested.

– On Figure 8. ‘ard’ is not explained.

We added “arrow-like depression” in the caption below Figure 8 to explain the “ard”.

3) Citation – reference is complete.– Change „Brandon and Carpenter 2005” to Kilbourne and Carpenter 2005– l. 1167: Sereno 1998 is Sereno 1999?

“Brandon and Carpenter 2005…” was cited by mistakes, thus it was deleted.

Line 1167: ‘Sereno 1998’ referred to ‘Sereno PC. 1998. A rationale for phylogenetic definitions, with application to the higher-level taxonomy of Dinosauria. Neues Jahrbuch für Geologie und Paläontologie-Abhandlungen 41-83. DOI: 10.1127/njgpa/210/1998/41’.

This citation was left out in the reference in the reviewed version, thus we added it in the revised one.

Reviewer #2 (Recommendations for the authors):I was delighted to be asked to review this manuscript after seeing the preprint circulating online a few months back. This is a very exciting specimen. Given how relatively abundant saurischian material is in the Lufeng and Fengjiahe formations, it was frustrating that the corresponding ornithischian record was so sparse! Finally China-and indeed Laurasian Asia!-has a good diagnostic thyreophoran from the Early Jurassic.

Thank you for the positive comment, we are also pleased to be bringing this new material to light!

In my public review, I mentioned that adding Yuxisaurus to a dataset with a broader sampling of basal non-thyreophoran ornithischian taxa (e.g., Han et al., 2018 – J. Syst. Palaeontol. Or a derivative dataset) would strengthen the study; however, I realize this would add a lot of work, and I don’t think it’s entirely necessary as you have convincingly demonstrated that the specimen is a thyreophoran, and the aim of your study was not to clarify character polarization in early ornithischian taxa.

Although this is an interesting suggestion we do not think it would add to the broader debate over ornithischian relationships for several different reasons. Firstly, *Yuxisaurus* is clearly a member of Thyreophora and a more deeply nested member of the clade than several other widely accepted thyreophoran taxa. Hence, its placement in Thyreophora is uncontroversial meaning that the most interesting question is what its relationships are within Thyreophora. To do this we focused on those analyses whose characters are most appropriate for estimating these relationships. By contrast, other broader ornithischian phylogenies do not include adequate character samples for more deeply nested thyreophoran clades (Ankylosauria and Stegosauria) and usually treat these clades as composite OTUs. Hence, these analyses would not add further to investigations of the position of *Yuxisaurus* within Thyreophora. Moreover, as we demonstrate that *Yuxisaurus* is not one of the earliest diverging thyreophorans it is highly unlikely to have any impact on the distribution of more generalised ornithischian characters. Finally, there are currently several competing hypotheses of ornithischian relationships (Boyd, Han, Dieudonné, silesaurs as ornithischians) and it’s not clear which of these should be used right now (and running all four would be a large amount of extra work). Given these reasons, and the referee’s own opinion that this exercise isn’t strictly necessary, we have opted not to conduct these further analyses as we don’t feel they would add any more substantive information to the paper.

I was able to download the two supplemental data files containing the modified phylogenetic matrices from Norman (2021) and Maidment et al., (2020). Although I was able to replicate the authors' results using the modified Norman matrix, as detailed below, Supplementary file 2 (the modified Maidment et al., dataset) differs from the matrix described in text and needs to be replaced with the correct matrix prior to publication.

Corrected.